# Deep Contract Design via Discontinuous Networks

**Tonghan Wang**
Harvard University
twang1@g.harvard.edu

**Paul Dütting**
Google Switzerland
duetting@google.com

**Dmitry Ivanov**
Israel Institute of Technology
divanov@campus.technion.ac.il

**Inbal Talgam-Cohen**
Israel Institute of Technology
italgam@cs.technion.ac.il

**David C. Parkes**
Harvard University[*]
parkes@eecs.harvard.edu

## Abstract

Contract design involves a *principal* who establishes contractual agreements about payments for outcomes that arise from the actions of an *agent*. In this paper, we initiate the study of deep learning for the automated design of optimal contracts. We introduce a novel representation: the *Discontinuous ReLU (DeLU) network*, which models the principal's utility as a discontinuous piecewise affine function of the design of a contract where each piece corresponds to the agent taking a particular action. DeLU networks implicitly learn closed-form expressions for the incentive compatibility constraints of the agent and the utility maximization objective of the principal, and support parallel inference on each piece through linear programming or interior-point methods that solve for optimal contracts. We provide empirical results that demonstrate success in approximating the principal's utility function with a small number of training samples and scaling to find approximately optimal contracts on problems with a large number of actions and outcomes.

## 1   Introduction

Contract theory studies the setting where a *principal* seeks to design a contract for rewarding an *agent* on the basis of the uncertain outcomes caused by the agent's private actions [8, 45]. Typical examples include a landlord who enters into a summer rental with a contract that includes penalties in the case of damage; a homeowner who engages a firm to complete a kitchen renovation with a contract that conditions payments on timely completion or functioning appliances; or an individual who employs a freelancer to do some design work with a contract that includes bonuses for completing the job.

A contract specifies payments to the agent, conditioned on outcomes. The principal is self-interested, with a value for each outcome and a cost for making payments. The agent is also self-interested, and responds to a contract by choosing an action that maximizes its expected utility (expected payment minus the cost of an action). The problem is to find a contract that maximizes the principal's utility (expected value minus expected payment), given that the agent will best respond to the contract. In economics, this is referred to as a problem of *moral hazard*, in that the agent is willing to privately act in its best interest given the contract (the "hazard" is that the behavior of the agent may be to the detriment of the principal).

The importance of contract design is evidenced by the 2016 Nobel Prize awarded to O. Hart and B. Holmström [51] and its broad application to real-world problems. Contract design is one of the three fundamental problems in the realm of economics involving asymmetric information and incentives, along with *mechanism design* [9] and *signalling* (Bayesian persuasion) [41]. However, while

---

[*]Also DeepMind, London UK

37th Conference on Neural Information Processing Systems (NeurIPS 2023).

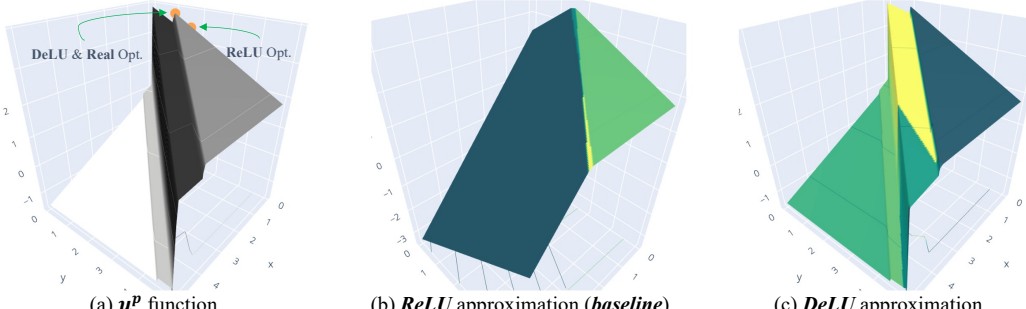

(a) $u^p$ function       (b) *ReLU* approximation (*baseline*)       (c) *DeLU* approximation

Figure 1: DeLU and ReLU approximation on a randomly generated contract design instance with 1,000 actions and 2 outcomes (see Sec. 4.3). The x- and y-coordinates are the payments for each of the two outcomes, and the z-coordinate is the utility of the principal. **(a)** The exact surface of the principal's utility function $u^p$. Different colors represent the action selected by the agent upon receiving the contract. **(b)** A learned *ReLU network* cannot model the discontinuity of the $u^p$ function and yields an incorrect contract as shown in (a). Colors in (b) and (c) represent different activation patterns (linear regions) of the networks. **(c)** A learned DeLU network represents a discontinuous function and can well-approximate $u^p$, yielding the optimal contract as shown in (a).

both mechanism design [10, 11, 12, 7, 36, 37] and signalling [26, 27, 18] have been studied extensively from a computational perspective, the contract design problem has only recently received attention and presented distinct computational challenges. For many combinatorial contract settings [6], the conventional approach based on linear programming becomes computationally infeasible and the problem of finding or even approximating the optimal contract becomes intractable [33, 28, 29]. In addition, learning-theoretic results give worst-case exponential sample complexity bounds for learning an approximately-optimal contract [40, 55]. As a result, there is a growing demand for a scalable, general purpose, and beyond-worst case approach for computing (near-)optimal contracts.

We are thus motivated to initiate the study of deep learning for optimal contract design (*deep contract design*). This falls within the broader framework of *differentiable economics*, which seeks to leverage parameterized representations of differentiable functions for the purpose of optimal economic design [30]. In regard to learning contracts, recent work [40, 21, 55, 31] considers the distinct setting of *online learning* for contract design with bandit feedback, characterizing regret bounds without appeal to deep learning.

A first innovation of this paper is to introduce a neural network architecture that can well-approximate the principal's utility function. A close examination of the geometry of the principal's utility function (Sec. 3) reveals its similarity to fully-connected feed-forward neural networks with ReLU activations [1]: both of them are piecewise affine functions [22]. However, whereas a ReLU network models a continuous function, the principal's utility function is discontinuous at the boundary of linear regions, where the best response of the agent changes. Accurate approximation in the vicinity of boundaries is critical because an optimal contract is always located on the boundary (Lemma 3 in Sec. 3), but this is exactly where ReLU approximation error can be large. To handle this, we introduce the *Discontinuous ReLU (DeLU)* network, which recognizes that the linear regions of a ReLU network are decided by activation patterns (the status of all activation units in the network), and conditions a *piecewise bias* on these activation patterns. In this way, each linear region has a different bias parameter and can be discontinuous at the boundaries. Fig. 1 illustrates the DeLU and its use to represent the principal's utility function, contrasting this with a continuous ReLU function.

A second innovation in this paper is to introduce a scalable inference technique that can find the contract that maximizes the network output (i.e., the principal's utility). We show that linear regions (pieces) of DeLU networks implicitly learn closed-form expressions for the incentive constraints of the agent and the utility maximization objective of the principal, so that we can use linear programming (LP) on each piece to find the global optimum. However, the time efficiency of this LP method is impeded by the overhead of solving individual LPs, and worsens with problem size in the absence of suitable parallel computing resources. To increase the computational efficiency, we develop a gradient-based inference algorithm based on the interior-point method [49] that only requires a few forward and backward passes of the DeLU network to find the contract that maximizes the network output. This method scales well to large-scale problems and can be readily run in parallel on GPUs.

The effectiveness of introducing piecewise discontinuity into neural networks is demonstrated by our experimental results. The DeLU network well-approximates the principal's utility function even with

a small number of training samples, and the inference method remains accurate and efficient as the problem size grows. By synergistically harnessing these two innovations, our method consistently finds near-optimal solutions on a wide range of contract design problems, significantly surpassing the capability of conventional continuous networks.

**Related work.** A longstanding challenge in the deep learning community has been to approximate discontinuous functions with neural networks. While the Universal Approximation Theorem guarantees the approximation of continuous functions, many problems involve discontinuity, including solar flare imaging [46] and problems coming from mathematics [24]. However, establishing a network to represent discontinuous functions is not an easy task. Discontinuities were considered as early as in the 1950s, when Rosenblatt [50] introduced the perceptron model and its single-layer of step activation functions. Following this work, others modeled discontinuity through the use of different discontinuous activation functions [34, 2, 44]. However, training these models is more challenging than the more typical models that make use of continuous activation functions [24], and this has hindered their application. To the best of our knowledge, this paper presents the first discontinuous network architecture with continuous activation functions and stable optimization performance.

There is a vast and still-growing economics literature on contracts [see, e.g., 52, 15], to which the computational lens has recently been applied. In classic settings, computing the optimal contract is tractable by solving one LP per agent's action, and taking the overall best solution [see, e.g., 32]. However, LP becomes computationally infeasible for combinatorial contracts, including settings with exponentially-many outcomes [33], actions [28], or agent combinations [29]. Another issue with the LP-based approach is that it requires complete information in regard to the problem facing the agent (its *type*). If the agent has a hidden type, this makes the optimal contract hard to compute [16, 38, 17], or otherwise complex [4]. One way to deal with these complexities is to focus on simple contracts, in particular *linear contracts* (commission-based), which are popular in practice [e.g., 13, 32, 14].

A distinct but related approach combines contract design with learning, where efforts have concentrated on online learning theory [40, 21, 55, 31]. These works show exponential lower bounds on the achievable regret for general contracts in worst-case settings, and sublinear regret bounds for linear contracts. Additionally, contracts have been studied from the perspective of multi-agent reinforcement learning [48, 19, 25]. The problem of *strategic classification* has also established a formal connection between strategy-aware classifiers and contracts [43, 42, 3]. There is also interest in the application of contract theory to the AI alignment problem [39]. However, the use of learning methods for solving optimal contracts, as opposed to auctions [30], remains largely unexplored. This gap in the literature can be partly attributed to the inherent complexity of finding optimal contracts, as the principal utility depends on the agent's action, which must conform to incentive compatibility (IC) constraints that are not observable by the principal.

## 2 Preliminaries

**Offline contract learning problem.** The contract design problem is defined with elements $\mathcal{C} = \langle \mathcal{A}, \mathcal{O}, \mathcal{F}, c, p, v \rangle$, and involves a single principal and a single agent. The agent selects an action $a$ in the finite action space $\mathcal{A}$, $|\mathcal{A}|=n$. Action $a$ leads to a distribution $p(\cdot|a)$ over the outcomes in $\mathcal{O}$, $|\mathcal{O}|=m$, and incurs a cost $c(a) \in \mathbb{R}_{\geq 0}$ to the agent. The valuation of each outcome $o_j \in \mathcal{O}$ for the principal is decided by the value function $v : \mathcal{O} \to \mathbb{R}_{\geq 0}$. The principal sets up a *contract*, $\boldsymbol{f} \in \mathcal{F} \subset \mathbb{R}_{\geq 0}^m$, which influences the action selected by the agent. A contract $\boldsymbol{f} = (f_1, f_2, \cdots, f_m)^\top$ specifies the payment $f_j \geq 0$ made to the agent by the principal in the event of outcome $o_j$. On receiving a contract $\boldsymbol{f}$, the agent selects the action $a^*(\boldsymbol{f})$ maximizing its utility $u^a(\boldsymbol{f}; a) = \mathbb{E}_{o \sim p(\cdot|a)}[f_o] - c(a)$. For a given action $a$ of the agent, the principal gets utility $u^p(\boldsymbol{f}; a) = \mathbb{E}_{o \sim p(\cdot|a)}[v_o - f_o]$, which is the principal's expected value minus payment. In the case that the induced action is the best response of the agent given the contract, we write $u^p(\boldsymbol{f}) = \mathbb{E}_{o \sim p(\cdot|a^*(\boldsymbol{f}))}[v_o - f_o]$. The goal in optimal contract design is to find the contract that maximizes the principal's utility without access to $p(\cdot|a)$ and the action taken by the agent:

$$\boldsymbol{f}^* = \arg\max_{\boldsymbol{f}} u^p(\boldsymbol{f}) = \arg\max_{\boldsymbol{f}} \mathbb{E}_{o \sim p(\cdot|a^*(\boldsymbol{f}))}[v_o - f_o]. \tag{1}$$

**ReLU piecewise-affine networks and activation patterns.** A fully-connected neural network with a piecewise linear activation function (e.g., ReLU and leaky ReLU) and a linear output layer represents a *continuous* piecewise affine function [5, 23]. A piecewise affine function is defined as follows.

**Definition 1** [Piecewise affine function]. *A function $g : \mathbb{R}^d \to \mathbb{R}$ is piecewise affine if there exists a finite set of polytopes $\{\mathcal{D}_i\}_{i=1}^P$ such that $\cup_{i=1}^P \mathcal{D}_i = \mathbb{R}^d$, $\mathcal{D}_i \cap \mathcal{D}_{j \neq i} = \varnothing$, and $g$ is a linear function $\rho_i : \mathcal{D}_i \to \mathbb{R}$ when restricted to $\mathcal{D}_i$. We call $\rho_i$ a* linear piece *of g.*

We now follow [22] and introduce some local properties of the piecewise affine function that is represented by a ReLU network. Suppose there are $L$ hidden layers in a network $g$, with sizes $[n_1, n_2, \cdots, n_L]$. $\boldsymbol{W}^{(l)} \in \mathbb{R}^{n_l \times n_{l-1}}$ and $\boldsymbol{b}^{(l)} \in \mathbb{R}^{n_l}$ are the weights and biases of layer $l$. Let $n_0 = d$ denote the input space dimension. We consider a ReLU network with one-dimensional outputs, and the output layer has weights $\boldsymbol{W}^{(L+1)} \in \mathbb{R}^{1 \times n_L}$ and a bias $b^{(L+1)} \in \mathbb{R}$. With input $\boldsymbol{x} \in \mathbb{R}^d$, we have the pre- and post-activation output of layer $l$: $\boldsymbol{h}^{(l)}(\boldsymbol{x}) = \boldsymbol{W}^{(l)} \boldsymbol{o}^{(l-1)}(\boldsymbol{x}) + \boldsymbol{b}^{(l)}$ and $\boldsymbol{o}^{(l)}(\boldsymbol{x}) = \sigma\left(\boldsymbol{h}^{(l)}(\boldsymbol{x})\right)$, where $\sigma : \mathbb{R} \to \mathbb{R}$ is an *activation function*. In this paper, we consider ReLU activation $\sigma(x) = \max\{x, 0\}$ [35, 54], but the proposed method can be extended to other piecewise linear activation functions (e.g., LeakyReLU and PReLU [54]). For each hidden unit, the ReLU *activation status* has two values, defined as 1 when pre-activation $h$ is positive and 0 when $h$ is strictly negative. The activation pattern of the entire network is defined as follows.

**Definition 2** [Activation Pattern]. *An* activation pattern *of a ReLU network g with L hidden layers is a binary vector $\boldsymbol{r} = [\boldsymbol{r}^{(1)}, \cdots, \boldsymbol{r}^{(L)}] \in \{0, 1\}^{\sum_{l=1}^L n_l}$, where $\boldsymbol{r}^{(l)}$ is a* layer activation pattern *indicating activation status of each unit in layer l.*

The activation pattern depends on the input $\boldsymbol{x}$, and we define function $r : \mathbb{R}^d \to \{0, 1\}^{\sum_{l=1}^L n_l}$ that maps the input to the corresponding activation pattern. For a ReLU network, inputs that have the same activation pattern lie in a polytope, and the activation pattern determines the boundaries of this polytope. To see this, we can write the output of layer $l$, $\boldsymbol{h}^{(l)}(\boldsymbol{x})$, as

$$\boldsymbol{h}^{(l)}(\boldsymbol{x}) = \boldsymbol{W}^{(l)} \boldsymbol{R}^{(l-1)}(\boldsymbol{x}) \left(\boldsymbol{W}^{(l-1)} \boldsymbol{R}^{(l-2)}(\boldsymbol{x}) \left(\cdots \left(\boldsymbol{W}^{(1)} \boldsymbol{x} + \boldsymbol{b}^{(1)}\right) \cdots\right) + \boldsymbol{b}^{(l-1)}\right) + \boldsymbol{b}^{(l)}, \quad (2)$$

where $\boldsymbol{R}^{(k)}$ is a diagonal matrix with diagonal elements equal to the layer activation pattern $\boldsymbol{r}^{(k)}$. Eq. 2 indicates that, when $\boldsymbol{r}$ is fixed, $\boldsymbol{h}^{(l)}$ is a linear function $\boldsymbol{h}^{(l)}(\boldsymbol{x}) = \boldsymbol{M}^{(l)} \boldsymbol{x} + \boldsymbol{z}^{(l)}$, where $\boldsymbol{M}^{(l)} = \boldsymbol{W}^{(l)} \left(\prod_{k=1}^{l-1} \boldsymbol{R}^{(l-k)}(\boldsymbol{x}) \boldsymbol{W}^{(l-k)}\right)$ and $\boldsymbol{z}^{(l)} = \boldsymbol{b}^{(l)} + \sum_{k=1}^{l-1} \left(\prod_{j=1}^{l-k} \boldsymbol{W}^{(l+1-j)} \boldsymbol{R}^{(l-j)}(\boldsymbol{x})\right) \boldsymbol{b}^{(k)}$. We thus get $\sum_{l=1}^L n_l$ half-spaces, with the half-space corresponding to unit $i$ of layer $l$ defined as:

$$\Gamma_{l,i} = \left\{\boldsymbol{y} \in \mathbb{R}^d | \Delta_i^{(l)} \left(\boldsymbol{M}_i^{(l)} \boldsymbol{y} + \boldsymbol{z}_i^{(l)}\right) \geq 0\right\}, \quad (3)$$

where $\boldsymbol{M}_i^{(l)} \boldsymbol{y} + \boldsymbol{z}_i^{(l)}$ is the output of unit $i$ at layer $l$, and $\Delta_i^{(l)}$ is 1 if $h_i^{(l)}(\boldsymbol{x})$ is positive, and is -1 otherwise. The input $\boldsymbol{x}$ is in the polytope that is defined by the intersection of these half-spaces: $\mathcal{D}(\boldsymbol{x}) = \cap_{l=1, \cdots, L} \cap_{i=1, \cdots, n_l} \Gamma_{l,i}$. When restricted to $\mathcal{D}(\boldsymbol{x})$, the ReLU network is a linear function: $g(\boldsymbol{x}) = \boldsymbol{W}^{(L+1)} \boldsymbol{R}^{(L)} \boldsymbol{h}^{(L)}(\boldsymbol{x}) + b^{(L+1)}$.

**Interior-point method for optimization problems with inequality constraints.** For a minimization problem with objective function $q(\boldsymbol{x})$ and inequality constraints $p_i(\boldsymbol{x}) > 0, i = 1, \cdots, M$, the *interior-point method* [53] introduces a logarithmic *barrier function*, $\phi(\boldsymbol{x}) = -\sum_{i=1}^M \log(p_i(\boldsymbol{x}))$, and finds the minimizer of $q(\boldsymbol{x}) + \frac{1}{t}\phi(\boldsymbol{x})$, for some $t > 0$. This new objective function is defined on the set of strictly feasible points $\{\boldsymbol{x} | p_i(\boldsymbol{x}) > 0, i = 1, \cdots, M\}$, and approximates the original objective as $t$ becomes large. Given this, we can solve for a series of optimization problems for increasing values of $t$. In the $k$-th round, $t^{(k)}$ is set to $\mu \cdot t^{(k-1)}$, where $\mu > 1$ is a constant, $t^{(0)} > 0$ is an initial value, and the problem is solved (e.g., by Newton initialized at $\boldsymbol{x}^{(k-1)}$) to yield $\boldsymbol{x}^{(k)}$. Assume that we solve the barrier problem exactly for each iterate, then to achieve a desired accuracy level of $\epsilon > 0$, we need $n_{\texttt{barrier}} = \log(M/(t^{(0)}\epsilon))/\log(\mu)$ rounds of optimization.

## 3  Geometry of Optimal Contracts

In this section, we introduce some properties of the principal's utility function, $u^p : \mathcal{F} \to \mathbb{R}$, which will motivate our method in Sec. 4. First, we show that $u^p$ is a piecewise affine function.

**Lemma 1.** *The principal's utility function $u^p$ is a piecewise affine function.*

*Proof.* The principal utility depends on the agent action. For contracts in the intersection of the following $n-1$ half spaces that represent *incentive compatibility constraints*, the agent's action is $a_i$:

$$\Gamma_{i,j} = \left\{ \boldsymbol{f} \in \mathcal{F} \mid \mathbb{E}_{o \sim p(\cdot | a_i)}\left[f_o\right] - c(a_i) \geq \mathbb{E}_{o \sim p(\cdot | a_j)}\left[f_o\right] - c(a_j) \right\}, \forall j \neq i. \tag{4}$$

Moreover, when agent action $a_i$ remains unchanged, $u^p$ changes linearly with $\boldsymbol{f}$: $\alpha u^p(\boldsymbol{f}; a_i) + \beta u^p(\boldsymbol{f}'; a_i) = \alpha \mathbb{E}_{o \sim p(\cdot | a_i)}\left[v_o - f_o\right] + \beta \mathbb{E}_{o \sim p(\cdot | a_i)}\left[v_o - f_o'\right] = \mathbb{E}_{o \sim p(\cdot | a_i)}\left[v_o - (\alpha f_o + \beta f_o')\right] = u^p(\alpha \boldsymbol{f} + \beta \boldsymbol{f}'; a_i)$. Therefore, when restricted to the polytope $\mathcal{Q}_i = \cap_{j \neq i} \Gamma_{i,j}, \forall i$, $u^p$ is linear. $\square$

Based on Lemma 1, we define a *linear piece* of function $u^p$ as $\mu_i^p : \mathcal{Q}_i \to \mathbb{R}$, where $\mathcal{Q}_i$ defines the set of contracts that motivates the agent to take action $i$. Lemma 1 can be easily extended to the agent's utility function $u^a$, and the linear pieces of function $u^p$ and $u^a$ share the same set of domains $\{\mathcal{Q}_i\}$. We define a linear piece of function $u^a$ as $\mu_i^a : \mathcal{Q}_i \to \mathbb{R}$. We next observe:

**Lemma 2.** *The principal's utility function $u^p$ can be discontinuous on the boundary of linear pieces.*

The proof in Appx. A follows the idea that the agent is indifferent between action $a_i$ and $a_j$ given a contract $\boldsymbol{f}$ on the boundary of two neighboring linear pieces $\mu_i^p$ and $\mu_{j \neq i}^p$: $\mu_i^a(\boldsymbol{f}) = \mu_j^a(\boldsymbol{f}) = u^a(\boldsymbol{f})$. However, the principal's utility equals the expected value minus the cost of an action minus $u^a(\boldsymbol{f})$. As the expected value minus the cost can be different in $\mu_i^p$ and $\mu_j^p$, $u^p$ can be discontinuous at $\boldsymbol{f}$.

We then define the optimality of a contract and analyze the structure of optimal contracts.

**Definition 3** [Piecewise and Global Optimal Contracts]**.** *A contract $\boldsymbol{f}_i^* \in \mathcal{Q}_i$ is piecewise optimal if $u^p(\boldsymbol{f}_i^*) \geq u^p(\boldsymbol{f}), \forall \boldsymbol{f} \in \mathcal{Q}_i$. A contract $\boldsymbol{f}^*$ is global optimal if $u^p(\boldsymbol{f}^*) \geq u^p(\boldsymbol{f}), \forall \boldsymbol{f} \in \mathcal{F}$.*

**Lemma 3.** *The global optimal contract is on the boundary of a linear piece.*

Lemma 3 can be proved by contradiction (as detailed in Appx. A): if a global optimal contract is not on the boundary, we can always find a solution with greater principal utility. As analyzed in Sec. 4, Lemma 3 serves as a compelling rationale for introducing our discontinuous networks. Besides discontinuity, there is another network design consideration motivated by the following property.

**Lemma 4.** *The principal's utility function $u^p$ can be written as a summation of a concave function and a piecewise constant function.*

The proof in Appx. A commences by establishing the convexity of function $u^a$ and subsequently formulating $u^p$ as a function of $u^a$. This property motivates us to introduce *concavity* into our network design. In Appx. C, we discuss how to achieve this by imposing non-negativity constrains on network weights and analyze the impact of this design choice on the performance.

## 4 DeLU Neural Networks

Given the piecewise-affine geometry, the preceding analysis shows a close connection between the principal's utility function $u^p$ (Lemma 1) and fully-connected ReLU networks (Sec. 2). However, ReLU functions are continuous and cannot represent the abrupt changes in $u^p$ at the boundaries of the linear pieces. This lack of representational capacity is problematic because the optimal contracts are on the boundary (Lemma 3), which is precisely where the ReLU approximation errors can be large. Consequently, ReLU networks are not well suited to deep contract design. In this section, we introduce the new *Discontinuous ReLU (DeLU) network*, which provides *a discontinuity at boundaries between pieces*, making it a suitable function approximator for the $u^p$ function.

### 4.1 Architecture

The DeLU network architecture supports different biases for different linear pieces. Since a linear piece can be identified by the corresponding activation pattern, we propose to condition these *piecewise biases* on activation patterns. Specifically, we learn a *DeLU network* $\xi : \mathcal{F} \to \mathbb{R}$ (Fig. 2) to approximate the principal's utility function, mapping a contract to the corresponding utility of the principal. The first part of a DeLU network is a sub-network similar to a conventional ReLU network. This sub-network $\eta$ has $L$ fully-connected hidden layers with ReLU activation and a weight matrix at the output layer, i.e., $\eta$ is parameterized as $\theta_\eta = \{\boldsymbol{W}^{(1)}, \boldsymbol{b}^{(1)}, \cdots, \boldsymbol{W}^{(L)}, \boldsymbol{b}^{(L)}, \boldsymbol{W}^{(L+1)}\}$, which includes weights and biases for $L$ hidden layers and weights for the output $((L+1)$-th) layer. The

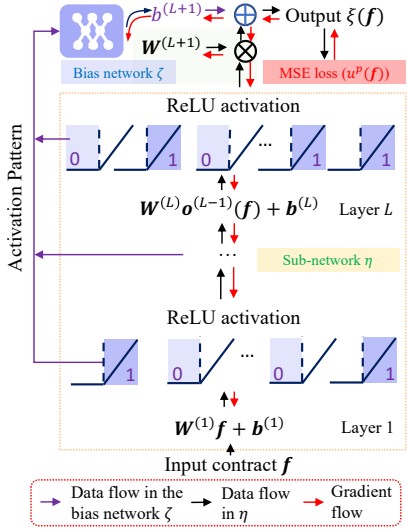

Figure 2: The DeLU architecture.

**Algorithm 1** Parallel Gradient-Based Inference

**Input:** $\boldsymbol{X}^{(0)} \in \mathbb{R}^{K \times m}$; DeLU network $\xi$ with trained parameters; $\epsilon$; $\mu > 1$; $t^{(0)}$; $\alpha$.

**for** $k \in \{1, \cdots, \lceil \frac{\log(N/t^{(0)}\epsilon)}{\log(\mu)} \rceil\}$ **do**
  $t^{(k)} = \mu^{k-1}t^{(0)}$; $\boldsymbol{X}^{(k,0)} \leftarrow \boldsymbol{X}^{(k-1)}$; /*Each round starts with the optimal solution in the last round.*/
  **for** $j = 1, 2, \cdots$ **do**
    $\boldsymbol{H}^{(l)} = \boldsymbol{M}_i^{(l)}\boldsymbol{X}^{(k,j-1)} + \boldsymbol{z}_i^{(l)}$ /*Get hidden layer outputs in a single forward pass.*/
    $\boldsymbol{\Phi}^{(k,j-1)} = -\sum_{l,i} \log \Delta_i^{(l)} \boldsymbol{H}_i^{(l)}$/*Barrier term.*/
    $d\boldsymbol{X} = \frac{\partial}{\partial \boldsymbol{X}^{(k,j-1)}} \left[ \boldsymbol{W}^{(L+1)}\boldsymbol{R}_i^{(L)}\boldsymbol{H}^{(L)} - \frac{\boldsymbol{\Phi}^{(k,j-1)}}{t^{(k)}} \right]$

    **if** $\|d\boldsymbol{X}\|_\infty < \epsilon$ **then** /*Converged at $t^{(k)}$*/
      $\boldsymbol{X}^{(k)} \leftarrow \boldsymbol{X}^{(k,j-1)}$; break; /*To the next round*/
    **else** $\boldsymbol{X}^{(k,j)} = \boldsymbol{X}^{(k,j-1)} + \alpha d\boldsymbol{X}$ /*Gradient ascent*/
  **end for**
**end for**

DeLU network is different from a conventional ReLU network at the bias of the output (last) layer $(b^{(L+1)})$, and we introduce a new method to generate the last-layer biases.

Since there are no activation units at the output layer, given an input contract $\boldsymbol{f}$, we can obtain the activation pattern $r(\boldsymbol{f})$ by a forward pass of the network up until the last layer. To condition $b^{(L+1)}$ on the activation pattern, we train another sub-network $\zeta : [0,1]^{\sum_{l=1}^L n_l} \to \mathbb{R}$, that maps $r(\boldsymbol{f})$ to the bias of the output layer. We use a two-layer fully-connected network with Tanh activation to represent $\zeta$ and denote its parameters as $\theta_\zeta$. In this way, contracts in the same linear piece of the DeLU network share the same bias value, enabling the network to express discontinuity at the boundaries while keeping other properties of network $\eta$ unchanged. In Appx. B, we discuss why we adopt a neural network, instead of a simpler function, to model the bias term.

### 4.2 Training and inference

The DeLU network $\xi$, including sub-networks $\eta$ and $\zeta$, is end-to-end differentiable. For training, we randomly sample $K$ contracts and the corresponding principal's utilities: $\mathcal{T}_K = \{(\boldsymbol{f}_i, u^p(\boldsymbol{f}_i))\}_{i=1}^K$. Feeding a training sample $\boldsymbol{f}_i$ as input, we get an approximated principal's utility: $\xi(\boldsymbol{f}_i; \theta_\eta, \theta_\zeta) = \eta(\boldsymbol{f}_i; \theta_\eta) + \zeta(r(\boldsymbol{f}_i); \theta_\zeta)$, and the network $\xi$ is trained to minimize the following loss function in $T$ epochs (Appx. D gives more details on network architecture, infrastructure, and training):

$$\mathcal{L}_{\mathcal{T}_K}(\theta_\eta, \theta_\zeta) = \frac{1}{K}\sum_{i=1}^K \left[\xi(\boldsymbol{f}_i; \theta_\eta, \theta_\zeta) - u^p(\boldsymbol{f}_i)\right]^2. \tag{5}$$

Given a trained DeLU network, $\xi : \mathcal{F} \to \mathbb{R}$, we have an approximation of the principal's utility function. The next step is to find a contract that maximizes the learned utility function, that is $\boldsymbol{f}^* = \arg\max_{\boldsymbol{f}} \xi(\boldsymbol{f})$. We call this the *inference process*. Since the function represented by a DeLU network is discontinuous, conventional first- or second-order optimization methods are not applicable, and, formally, we need to develop a suitable optimization approach to solve

$$\xi(\boldsymbol{f}^*) = \max_{\rho_i} \max_{\boldsymbol{f} \in \mathcal{D}_i} \rho_i(\boldsymbol{f}), \tag{6}$$

where $\rho_i : \mathcal{D}_i \to \mathbb{R}$ is a linear piece of network $\xi$. This motivates us to first find the piecewise optimal contracts and then get the global optimum by comparing the piecewise optima.

#### 4.2.1 Linear programming based inference

A first approach finds the optimal contract for each piece using linear programming (LP). Contracts in the same linear piece $\rho$ result in the same activation pattern $r$, and the DeLU network is a linear function on the piece. In particular, the optimization problem of finding the piecewise optimum is:

$$\arg\max_{\boldsymbol{f}} \quad \boldsymbol{W}^{(L+1)}\boldsymbol{R}^{(L)}\left[\boldsymbol{M}^{(L)}\boldsymbol{f} + \boldsymbol{z}^{(L)}\right] \quad s.t. \ \Delta_i^{(l)}\left(\boldsymbol{M}_i^{(l)}\boldsymbol{f} + \boldsymbol{z}_i^{(l)}\right) \geq 0, \forall l \in [L], \ i \in [n_l],$$

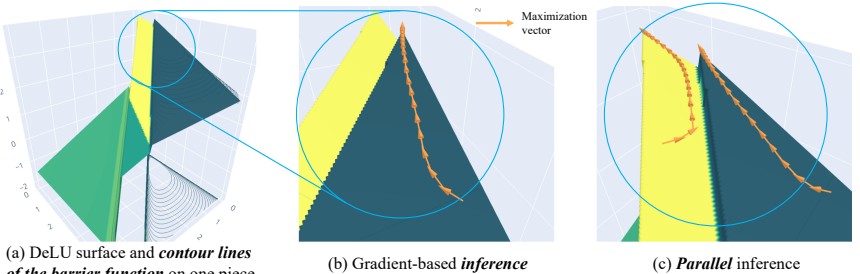

(a) DeLU surface and *contour lines of the barrier function* on one piece.

(b) Gradient-based *inference*

(c) *Parallel* inference

Figure 3: The gradient-based inference method for finding piecewise optima, illustrated on the same instance as Fig. 1.

where $[L] = \{1, \cdots, L\}$, $[n_l] = \{1, \cdots, n_l\}$, the objective is the linear function represented by the DeLU network on piece $\rho$, and the constraints require that the activation pattern remains $r$. The definitions of $\Delta_i^{(l)}$, $R^{(L)}$, $M^{(l)}$, and $z^{(l)}$ are the same as in Sec. 2, but with the activation pattern fixed to $r$. We omit the last-layer bias in the objective because the activation pattern is fixed for this piece, and thus the bias is constant and does not influence the piecewise optimal solution.

For each LP, there are $m$ decision variables, representing payments for $m$ outcomes, and $N = \sum_{l=1}^{L} n_l$ (the number of neurons) constraints. LPs for each piece can be solved in polynomial time of $m$ and $N$, and quickly in practice via the simplex method. However, a challenge is that there are $2^N$ activation patterns.[1] For a DeLU network with a moderate size, we can enumerate all these pieces. For a larger DeLU network, we can approximate by collecting random contract samples and solving LPs with the corresponding activation patterns. Parallelizing these LPs may achieve better time efficiency. However, this improvement is limited by the overhead of solving a single LP and the required amount of suitable parallel computational resources. In the next section, we introduce a gradient-based method that provides better efficiency in solving single problems and better parallelism through making use of GPUs. We compare these two inference methods in Sec. 5.1.

### 4.2.2   Gradient-based inference

The gradient-based inference method is based on the interior-point method (Sec. 2) and finds piecewise optimal solutions via forward and backward passes of the DeLU network. Specifically, on each linear piece, $\rho$, we adopt the following *barrier function*:

$$\phi(\boldsymbol{f}) = -\sum_{l=1}^{L} \sum_{i=1}^{n_l} \log \left[ \Delta_i^{(l)} \left( \boldsymbol{M}_i^{(l)} \boldsymbol{f} + \boldsymbol{z}_i^{(l)} \right) \right]. \tag{7}$$

At the $k$-th round, the objective function is

$$\phi^{(k)}(\boldsymbol{f}) = -\boldsymbol{W}^{(L+1)} \boldsymbol{R}^{(L)} \left[ \boldsymbol{M}^{(L)} \boldsymbol{f} + \boldsymbol{z}^{(L)} \right] + \frac{1}{t^{(k)}} \phi(\boldsymbol{f}). \tag{8}$$

We use gradient descent initialized at $\boldsymbol{f}^{(k-1)}$, and update $\boldsymbol{f}$ with $\partial \phi^{(k)}(\boldsymbol{f})/\partial \boldsymbol{f}$ to find the minimizer. A forward pass of the DeLU network gives $\phi^{(k)}(\boldsymbol{f})$ and a backward pass is sufficient to calculate $\partial \phi^{(k)}(\boldsymbol{f})/\partial \boldsymbol{f}$. Since forward and backward passes are naturally parallelized in modern deep learning frameworks, this method can be parallelized by processing multiple $\boldsymbol{f}$ simultaneously. Alg. 1 gives the matrix-form expression of this parallel computation. The input $\boldsymbol{X}^{(0)} \in \mathbb{R}^{K \times m}$ can be the training set or a random sample set, and we discuss the difference of these two settings in Appx. E.

**Inference performance and boundary alignment degree.** The performance of the proposed inference algorithms depends on the DeLU approximation quality, especially on the degree of alignment between the DeLU boundaries and the ground-truth linear piece boundaries. An interesting question is whether our training setup can achieve a high *boundary alignment degree*. A reason to think this is possible comes from observing that the MSE training loss (Eq. 5) is sensitive to misalignment between the DeLU and true boundaries. In particular, given that the jump of the utility function at boundary points can be arbitrarily large, a slight misalignment between DeLU and true boundaries can lead to a large increase in the MSE loss. Related to this, we explore an extension of the gradient-based inference method to make it more robust to possible boundary misalignment. When

---

[1]Previous research [20] found some patterns are invalid, and there are actually $n_{N,m} = \sum_{i=0}^{m} \binom{N}{m-i}$ pieces.

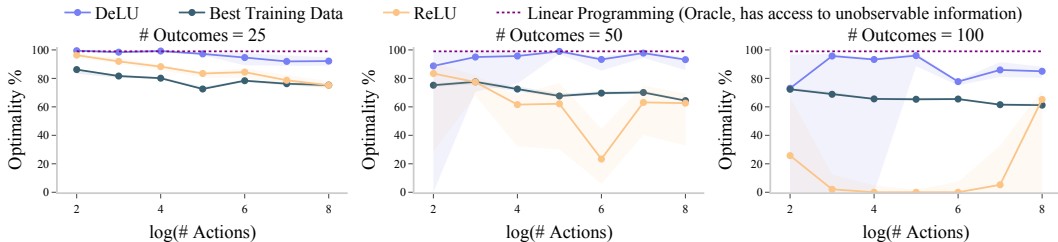

Figure 4: Optimality (normalized principal utility) of DeLU, ReLU and a direct LP solver (`Oracle LP`), for problems with increasing sizes.

annealing the coefficient of the barrier function, we can check whether the principal's utility increases for each $t^{(k)}$ value. A non-increasing utility indicates that we encounter inaccurate boundaries, and we can stop the inference to seek more robustness. We call inference with this "early stop" mechanism the *sub-argmax gradient-based inference method*.

In Sec. 5.3, we empirically evaluate the boundary alignment degree achieved by DeLU, and demonstrate that near-optimal contract-design performance is closely associated with high boundary alignment. We also show that the sub-argmax variation can improve performance of gradient-based inference, especially on tasks where the boundary alignment degree is low.

### 4.3 Illustrating DeLU-based contract design

We first illustrate the DeLU approach on a randomly generated contract design instance, in this case with 2 outcomes and 1,000 actions. In this instance, we sample the values for outcomes and costs for actions uniformly from $[0, 10]$. The outcome distribution for an action is further generated by applying `SoftMax` to a Gaussian random vector in $\mathbb{R}^m$. In Fig. 1 (a), we show the exact surface of the principal's utility function $u^p$ on such a randomly generated instance. We observe that $u^p$ is a discontinuous, piecewise affine function, where each piece corresponds to an action of the agent. In Fig. 1 (b) and (c), we show the $u^p$ surface approximated by each of a ReLU and a DeLU network trained with $40K$ samples. These two networks have a similar architecture, with a single hidden layer of 8 ReLU units. The difference is the additional, last-layer biases of the DeLU network, which are dependent on activation patterns. Whereas the ReLU network cannot represent the discontinuity of $u^p$, and thus gives an incorrect optimal contract, the DeLU network replicates the $u^p$ surface, and gives an accurate optimal contract (Fig. 1 (a)). Another interesting observation is that the DeLU network may use multiple pieces to represent an original linear region (Fig. 1 (c)). In Fig. 3, we further illustrate the inference process of the gradient-based method on this instance.

## 5 Empirical Evaluation

In this section, we design experiments to study the following aspects of DeLU contract design: **(1) Optimality** (Sec. 5.1): Can DeLU networks give solutions close to the optimal contracts? How do the solutions compare against those generated by continuous neural networks? **(2) Sample efficiency** (Sec. 5.1): How many training samples are required for accurate DeLU approximation? Is the proposed method applicable to large-scale problems? **(3) Time efficiency** (Sec. 5.2): How does the computation overhead required for DeLU learning and inference compare to those of other solvers? **(4) Inference** (Sec. 5.2): Does the gradient-based inference method provide a good tradeoff between accuracy and time efficiency? **(5) Boundary alignment degree** (Sec. 5.3): How does the boundary alignment degree affect optimality?

**Problem generation.** Experiments are carried out on random synthetic examples. The outcome distributions $p(\cdot|a)$ are generated by applying `SoftMax` on a Gaussian random vector in $\mathbb{R}^m$. The outcome value $v_o$ is uniform on $[0, 10]$. The action cost is a mixture, $c(a) = (1 - \beta_p)c_r(a) + \beta_p c_i(a)$, where $c_r(a) = \alpha_p \mathbb{E}_{o \sim p(\cdot|a)}[v_o]$ for scaling factor $\alpha_p > 0$ is a correlated cost that is proportional to the expected value of the action, $c_i(a)$ is an independent cost and uniform on $[0, 1]$, and $\beta_p$ controls the weight of the independent cost. We test different problem sizes by changing the number of outcomes $m$ and actions $n$. For each problem size, we test various combinations of $(\alpha_p, \beta_p)$ to consider the influence of correlation costs. Different methods are compared on the same set of problems.

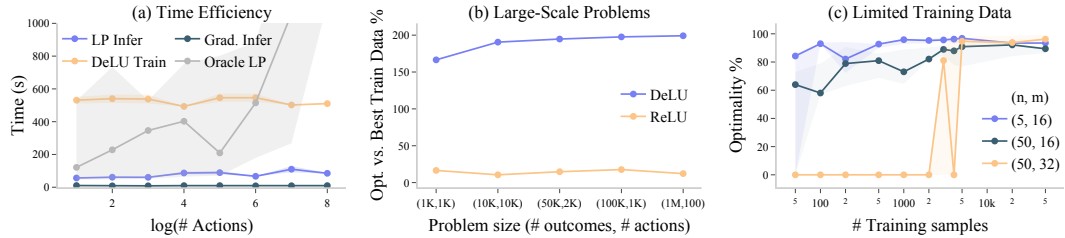

Figure 5: (a) DeLU training and inference time compared against `Oracle` LP. (b) DeLU performance on large-scale problems. (c) DeLU performance with a small number of training samples.

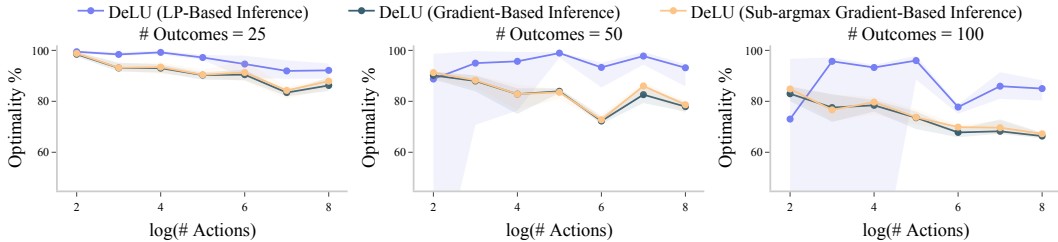

Figure 6: Comparing the optimality of the two inference methods for solving the global optimum given a learned DeLU network, considering increasing problem sizes.

## 5.1 Optimality and efficiency.

In Fig. 4, we compare DeLU against ReLU networks as well as a baseline *linear programming (LP)* solver (`Oracle` LP). `Oracle` LP refers to the use of LP for directly solving the contract design problem, not for inference on a trained DeLU network. It solves $n$ LP problems, one for each action $a$, where the objective is to maximize $u^p$ with the incentive compatibility (IC) constraints associated with the action $a$. The best of these $n$ solutions becomes the optimal contract. `Oracle` LP has access to the outcome distributions $p(\cdot|a)$ (to construct the IC constraints) that are unobservable to the DeLU and ReLU learners, but gives a *benchmark* for the optimality of the proposed method.

We test different problem sizes in Fig. 4. Specifically, we set the number of outcomes $m$ to 25 (1st column), 50 (2nd column), and 100 (3rd column) and increase the number of actions $n$ from $2^2$ to $2^8$. For each problem size, 12 combinations of $\alpha_p$ and $\beta_p$ are tested, with $(\alpha_p, \beta_p) \in \{0.5, 0.7, 0.9\} \times \{0, 0.3, 0.6, 0.9\}$. The median performance as well as the first and third quartile (shaped area) of these 12 combinations are shown. When reporting optimality, we normalize the principal's utility achieved by DeLU/ReLU LP-based inference via dividing them by the value returned by `Oracle` LP. To ensure a fair comparison, DeLU and ReLU networks have the same architecture, with one hidden layer of 32 hidden units. They are trained for 100 epochs with $50K$ random samples.

DeLU consistently achieves better **optimality** than ReLU networks ($+28.29\%$) and the best contract in the training set ($+23.84\%$) across all problem sizes. The performance gap is particularly large for large-scale problems. For example, when the number of outcomes is larger than $1,000$ (Fig. 5 (b)), the ReLU networks return solutions much worse ($< 20\%$) than training samples, while DeLU can obtain a solution at least $1.7$ times better than the best training data. As for **sample efficiency**, as shown in Fig. 5 (c), DeLU achieves near-optimality even with a very small training set.

## 5.2 Comparing the two DeLU inference methods.

In Fig. 6, we fix $m$ to 25, 50, 100 and increase $n$ from $2^2$ to $2^8$ to compare the optimality of the proposed inference methods. We again test the same 12 combinations of $(\alpha_p, \beta_p)$. The median performance and the first and third quartile are shown for each problem size. Gradient-based inference is parallelized for $50K$ training samples on GPUs, while LP inference is parallelized for 5 linear pieces on CPUs. We can see that LP inference consistently provides a better solution, as the optimality of gradient-based inference ($-7.56\%$) is limited by the number of gradient descent steps.

The advantage of gradient-based inference is its time efficiency. In Fig. 5 (a), we compare the inference time for different problem sizes (with $m$ fixed to 25). Gradient-based inference saves around $50$-$90\%$ overhead compared to LP inference. We also note that the DeLU training and

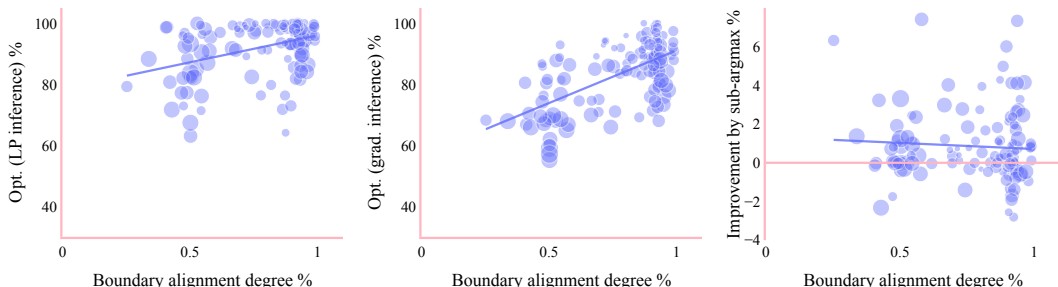

Figure 7: Correlation between boundary alignment degree (x-axis) and DeLU optimality (y-axis; left: LP-based inference; middle: gradient-based inference); also, the improvement achieved by sub-argmax inference compared to gradient-based inference (right). Trend lines (linear regression) are shown. Point sizes ($\log(mn)$) indicate the corresponding problem size.

inference costs do not increase with the problem size. By comparison, the overhead of the direct LP solver grows quickly with the problem scale.

## 5.3  Boundary alignment degree and its influence on DeLU optimality

In Fig. 7, we study the relationship between DeLU performance and the degree of boundary alignment.

For each contract design problem, we first calculate the boundary alignment degree achieved by the DeLU network. For this, we test a large number (50K) of contracts, and check whether they are simultaneously on the DeLU model and true utility boundary. Specifically, for each contract we randomly sample 10K directions and assess linearity of the DeLU utility model and the true utility function in each direction. The principal's utility function is piecewise linear in the proximity of an interior point. Conversely, when a point lies on a boundary, there is a jump in utility within its proximity, rendering the utility unable to pass a linearity test in some direction. In particular, if a function is non-linear in >20% of random directions, we mark the contract sample as being on a boundary (we use the exact same approach to check for boundaries of the DeLU utility function and the principal's true utility function). The *boundary alignment degree* is calculated as the percentage of overlapped boundary points (# contract samples on both DeLU and true boundaries / # contract samples on true boundaries).

Each point in Fig. 7 represents a contract design problem, and we use the same set of problems as in the previous experiments. The x-axis is the DeLU boundary alignment degree, and the y-axis is the optimality of DeLU with LP-based inference (Fig. 7 left) and gradient-based inference (Fig. 7 middle). The right plot gives the optimality improvement achieved by the sub-argmax inference method compared to standard gradient-based inference. From Fig. 7 left, we observe that DeLU achieves good boundary alignment degrees (>80%) for most contract design problems, and that a strong positive correlation exists between the boundary alignment degree and the optimality of LP-based inference. This result indicates that the alignment degree between DeLU and true boundaries is important in supporting good performance of LP-based inference. A similar observation can be made for gradient-based inference. Fig. 7-right shows that as the boundary alignment degree increases, the optimality improvement from the sub-argmax inference compared to standard gradient-based inference becomes less significant. This confirms that the sub-argmax inference is especially helpful for contract design problems where the DeLU boundaries are less accurate.

## 6  Closing Remarks

This paper initiates the investigation of contract design from a deep learning perspective, introducing a family of piecewise discontinuous networks and inference techniques that are tailored for deep contract learning. In future work, it will be interesting to take this framework to real-world settings, provide theory in regard to the expressiveness of the function class comprising these piecewise discontinuous functions, and extend to the setting of online learning. We expect that our exploration of discontinuous networks can also draw attention to other economics problems involving discontinuity and thereby contribute to advancing AI progress in computational economics.

## 7 Acknowledgements

This work received funding from the European Research Council (ERC) under the European Union's Horizon 2020 research and innovation program (grant agreement: 101077862, project name ALGO-CONTRACT). We extend our heartfelt appreciation to the anonymous NeurIPS reviewers for their insightful questions and constructive interactions during the review process, which has inspired us to delve deeper into critical inquiries, such as the boundary alignment issue. Their feedback has been important in shaping the quality and depth of our work.

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

# A  Proofs of Lemma 2-4

In Sec. 3, we study the geometric structure of optimal contracts by establishing four lemmas. While some of them have been stated and proved for linear contracts [32], and the first two lemmas, at least, are not surprising, we give formal proofs of these properties of the principal's utility and optimal contracts in the general case. An important property of the principal's utility function $u^p$ that we state in Lemma 2 is that $u^p$ can be a discontinuous function.

**Lemma 2.** *The principal's utility function $u^p$ can be discontinuous on the boundary of linear pieces.*

*Proof.* For a contract $\boldsymbol{f}$ on the boundary of two neighboring linear pieces $\mu_i^p$ and $\mu_{j\neq i}^p$, the agent is indifferent between action $a_i$ and $a_j$ given $\boldsymbol{f}$: $\mu_i^a(\boldsymbol{f}) = \mu_j^a(\boldsymbol{f})$. The principal's utility

$$\mu_i^p(\boldsymbol{f}) = \mathbb{E}_{o\sim p(\cdot|a_i)}[v_o] - \mathbb{E}_{o\sim p(\cdot|a_i)}[f_o] \tag{9}$$

$$= \mathbb{E}_{o\sim p(\cdot|a_i)}[v_o] - c(a_i) - (\mathbb{E}_{o\sim p(\cdot|a_i)}[f_o] - c(a_i)) \tag{10}$$

$$= \mathbb{E}_{o\sim p(\cdot|a_i)}[v_o] - c(a_i) - \mu_i^a(\boldsymbol{f}) \tag{11}$$

$$= \mathbb{E}_{o\sim p(\cdot|a_i)}[v_o] - c(a_i) - \mu_j^a(\boldsymbol{f}) \tag{12}$$

$$= \mu_j^p(\boldsymbol{f}) + \mathbb{E}_{o\sim p(\cdot|a_i)}[v_o] - c(a_i) - (\mathbb{E}_{o\sim p(\cdot|a_j)}[v_o] - c(a_j)). \tag{13}$$

It is possible that

$$\mathbb{E}_{o\sim p(\cdot|a_i)}[v_o] - c(a_i) \neq \mathbb{E}_{o\sim p(\cdot|a_j)}[v_o] - c(a_j). \tag{14}$$

Function $u^p$ is discontinuous in this case. $\square$

Another property of the principal's utility function $u^p$ that motivates our discontinuous neural networks is Lemma 3.

**Lemma 3.** *The global optimal contract is on the boundary of a linear piece.*

*Proof.* Suppose that the global optimal $\boldsymbol{f}^*$ is not on a boundary but is an interior point of a linear piece $\mathcal{Q}_i$ (contracts in $\mathcal{Q}_i$ encourage the agent to take action $a_i$.):

$$\boldsymbol{f}^* \in \mathcal{Q}_i - \partial\mathcal{Q}_i, \tag{15}$$

where

$$\mathcal{Q}_i = \cap_{j\neq i}\Gamma_{i,j}; \tag{16}$$

$$\Gamma_{i,j} = \left\{\boldsymbol{f} \in \mathcal{F} \mid \mathbb{E}_{o\sim p(\cdot|a_i)}[f_o] - c(a_i) \geq \mathbb{E}_{o\sim p(\cdot|a_j)}[f_o] - c(a_j)\right\}, \forall j \neq i. \tag{17}$$

It follows that

$$\boldsymbol{f}^* \in \left\{\boldsymbol{f} \in \mathcal{F} \mid \mathbb{E}_{o\sim p(\cdot|a_i)}[f_o] - c(a_i) > \mathbb{E}_{o\sim p(\cdot|a_j)}[f_o] - c(a_j), \forall j \neq i\right\}, \tag{18}$$

where the inequality is strict. Let

$$\epsilon_{i,j} = \mathbb{E}_{o\sim p(\cdot|a_i)}[f_o^*] - c(a_i) - \mathbb{E}_{o\sim p(\cdot|a_j)}[f_o^*] + c(a_j). \tag{19}$$

It holds that $\epsilon_{i,j} > 0, \forall j \neq i$.

Now we consider the contract $\boldsymbol{f}' = \boldsymbol{f}^* - \delta p(\cdot|a_i)$ for some small $\delta > 0$. For any $a_j \neq a_i$, we have

$$u^a(\boldsymbol{f}'; a_i) - u^a(\boldsymbol{f}'; a_j)$$
$$= \mathbb{E}_{o\sim p(\cdot|a_i)}[f_o^* - \delta p(o|a_i)] - c(a_i) - \mathbb{E}_{o\sim p(\cdot|a_j)}[f_o^* - \delta p(o|a_i)] + c(a_j) \tag{20}$$
$$= \epsilon_{i,j} - \delta\mathbb{E}_{o\sim p(\cdot|a_i)}[p(o|a_i)] + \delta\mathbb{E}_{o\sim p(\cdot|a_j)}[p(o|a_i)].$$

When

$$0 < \delta < \min_{j\neq i} \frac{\epsilon_{i,j}}{\mathbb{E}_{o\sim p(\cdot|a_i)}[p(o|a_i)] - \mathbb{E}_{o\sim p(\cdot|a_j)}[p(o|a_i)]}, \tag{21}$$

(where note the denominator is $> 0$), we have

$$u^a(\boldsymbol{f}'; a_i) - u^a(\boldsymbol{f}'; a_j) > 0, \forall j \neq i, \tag{22}$$

which means $\boldsymbol{f}'$ incentivizes the agent to take action $a_i$. Therefore, for $\delta$ in this range, the principal's utility given $\boldsymbol{f}'$ is:

$$
\begin{aligned}
u^p(\boldsymbol{f}') &= \mathbb{E}_{o\sim p(\cdot|a_i)}\left[v_o - f_o^* + \delta p(o|a_i)\right] \\
&= \mathbb{E}_{o\sim p(\cdot|a_i)}\left[v_o - f_o^*\right] + \mathbb{E}_{o\sim p(\cdot|a_i)}\left[\delta p(o|a_i)\right] \\
&= u^p(\boldsymbol{f}^*) + \delta \sum_o p^2(o|a_i) \\
&> u^p(\boldsymbol{f}^*).
\end{aligned}
\tag{23}
$$

We thus find a contract $\boldsymbol{f}'$ that induces greater utility for the principal, contradicting with the fact that $\boldsymbol{f}^*$ is the global optimal contract. This finishes the proof.

Note that here we consider the boundary resulting from changes in the agent's best responses. We can extend the proof to cover another type of boundary, which pertains to the requirement that contracts are non-negative, by defining $\mathcal{Q}_i$ to be $\mathcal{Q}_i = \{\boldsymbol{f}|\boldsymbol{f} \geq 0\} \cap \Gamma_{i,1} \cap \cdots \cap \Gamma_{i,i-1} \cap \Gamma_{i,i+1} \cap \cdots$. $\qquad\square$

Lemma 4 claims another property that may influence the design of DeLU networks.

**Lemma 4.** *The principal's utility function $u^p$ can be written as a summation of a concave function and a piecewise constant function.*

*Proof.* We first prove the agent's utility function $u^a(\boldsymbol{f})$ is a convex function.

We need to prove that, for any two contracts $\boldsymbol{f}^{(1)} \in \mathcal{F}$ and $\boldsymbol{f}^{(2)} \in \mathcal{F}$, it holds that $u^p(\lambda \boldsymbol{f}^{(1)} + (1 - \lambda)\boldsymbol{f}^{(2)}) \leq \lambda u^p(\boldsymbol{f}^{(1)}) + (1 - \lambda)u^p(\boldsymbol{f}^{(2)}), \forall \lambda \in [0, 1]$. Denote $\boldsymbol{d} = \boldsymbol{f}^{(2)} - \boldsymbol{f}^{(1)}$. It suffices to prove that the derivative of $u^p(\boldsymbol{f}^{(1)} + \delta \boldsymbol{d})$ with respect to $\delta$ is a non-decreasing function for $\delta \in [0, 1]$.

We have

$$
\frac{\partial}{\partial \delta}u^a(\boldsymbol{f}^{(1)} + \delta \boldsymbol{d}) = \frac{\partial}{\partial \delta}\left[\mathbb{E}_{o\sim p(\cdot|a_\delta)}[f_o^{(1)} + \delta d_o] - c(a_\delta)\right],
\tag{24}
$$

where $a_\delta$ is the agent's action given the contract $\boldsymbol{f}^{(1)} + \delta \boldsymbol{d}$.

Case 1: When $a_\delta$ does not change,

$$
\frac{\partial}{\partial \delta}u^a(\boldsymbol{f}^{(1)} + \delta \boldsymbol{d}) = \mathbb{E}_{o\sim p(\cdot|a_\delta)}[d_o]
\tag{25}
$$

is a constant, which is a non-decreasing function.

Case 2: When $a_\delta$ changes. Suppose that there exists $\delta_1 \in [0, 1]$ such that $\boldsymbol{f}^{(1)} + \delta_1 \boldsymbol{d}$ is on the boundary of linear piece $\mathcal{Q}_i$ and $\mathcal{Q}_j$. Let

$$
\begin{aligned}
\delta_1^+ &= \delta_1 + \epsilon, \\
\delta_1^- &= \delta_1 - \epsilon,
\end{aligned}
\tag{26}
$$

where $\epsilon$ is a small number and

$$
\begin{aligned}
\boldsymbol{f}^{(1)} + \delta_1^- \boldsymbol{d} &\in \mathcal{Q}_i, \\
\boldsymbol{f}^{(1)} + \delta_1^+ \boldsymbol{d} &\in \mathcal{Q}_j.
\end{aligned}
\tag{27}
$$

Because the agent is self-interested, it follows that $a_j$ is the best response when the contract is $\boldsymbol{f}^{(1)} + \delta_1^+ \boldsymbol{d}$:

$$
u^a(\boldsymbol{f}^{(1)} + \delta_1^+ \boldsymbol{d}) = u^a(\boldsymbol{f}^{(1)} + \delta_1^+ \boldsymbol{d}; a_j) > u^a(\boldsymbol{f}^{(1)} + \delta_1^+ \boldsymbol{d}; a_i).
\tag{28}
$$

It follows that

$$
\frac{\partial}{\partial \delta}u^a(\boldsymbol{f}^{(1)} + \delta \boldsymbol{d})\bigg|_{\delta=\delta_1^+} = \lim_{\epsilon\to 0}\frac{u^a(\boldsymbol{f}^{(1)} + \delta_1^+ \boldsymbol{d}) - u^a(\boldsymbol{f}^{(1)} + \delta_1 \boldsymbol{d})}{\epsilon}
\tag{29}
$$

$$
> \lim_{\epsilon\to 0}\frac{u^a(\boldsymbol{f}^{(1)} + \delta_1^+ \boldsymbol{d}; a_i) - u^a(\boldsymbol{f}^{(1)} + \delta_1 \boldsymbol{d})}{\epsilon}.
\tag{30}
$$

We now look at the two terms in the numerator of Eq. 30:

$$u^a(\boldsymbol{f}^{(1)} + \delta_1^+ \boldsymbol{d}; a_i) = \mathbb{E}_{o \sim p(\cdot|a_i)}[f_o^{(1)} + \delta_1^+ d_o] - c(a_i) \tag{31}$$

$$= \mathbb{E}_{o \sim p(\cdot|a_i)}[f_o^{(1)} + (\delta_1 + \epsilon)d_o] - c(a_i) \tag{32}$$

$$= \mathbb{E}_{o \sim p(\cdot|a_i)}[f_o^{(1)} + \delta_1 d_o] - c(a_i) + \epsilon \mathbb{E}_{o \sim p(\cdot|a_i)}[d_o] \tag{33}$$

$$= u^a(\boldsymbol{f}^{(1)} + \delta_1 \boldsymbol{d}) + \epsilon \mathbb{E}_{o \sim p(\cdot|a_i)}[d_o], \tag{34}$$

and

$$u^a(\boldsymbol{f}^{(1)} + \delta_1 \boldsymbol{d}) = \mathbb{E}_{o \sim p(\cdot|a_i)}[f_o^{(1)} + \delta_1 d_o] - c(a_i) \tag{35}$$

$$= \mathbb{E}_{o \sim p(\cdot|a_i)}[f_o^{(1)} + (\delta_1^- + \epsilon)d_o] - c(a_i) \tag{36}$$

$$= \mathbb{E}_{o \sim p(\cdot|a_i)}[f_o^{(1)} + \delta_1^- d_o] - c(a_i) + \epsilon \mathbb{E}_{o \sim p(\cdot|a_i)}[d_o] \tag{37}$$

$$= u^a(\boldsymbol{f}^{(1)} + \delta_1^- \boldsymbol{d}) + \epsilon \mathbb{E}_{o \sim p(\cdot|a_i)}[d_o]. \tag{38}$$

Therefore,

$$\frac{\partial}{\partial \delta} u^a(\boldsymbol{f}^{(1)} + \delta \boldsymbol{d})\bigg|_{\delta = \delta_1^+} > \lim_{\epsilon \to 0} \frac{u^a(\boldsymbol{f}^{(1)} + \delta_1^+ \boldsymbol{d}; a_i) - u^a(\boldsymbol{f}^{(1)} + \delta_1 \boldsymbol{d})}{\epsilon}$$

$$= \lim_{\epsilon \to 0} \frac{u^a(\boldsymbol{f}^{(1)} + \delta_1 \boldsymbol{d}) - u^a(\boldsymbol{f}^{(1)} + \delta_1^- \boldsymbol{d})}{\epsilon} \tag{39}$$

$$= \frac{\partial}{\partial \delta} u^a(\boldsymbol{f}^{(1)} + \delta \boldsymbol{d})\bigg|_{\delta = \delta_1^-}.$$

This finishes the proof that $u^a$ is a convex function. Furthermore, we have that

$$u^p(\boldsymbol{f}) = -u^a(\boldsymbol{f}) - c(a^*(\boldsymbol{f})) + \mathbb{E}_{o \sim p(\cdot|a^*(\boldsymbol{f}))}[v_o], \tag{40}$$

where $-u^a(\boldsymbol{f})$ is a concave function and $-c(a^*(\boldsymbol{f})) + \mathbb{E}_{o \sim p(\cdot|a^*(\boldsymbol{f}))}[v_o]$ is a piecewise constant function with the value $-c(a_i) + \mathbb{E}_{o \sim p(\cdot|a_i)}[v_o]$ when $\boldsymbol{f} \in \mathcal{Q}_i$. $\qquad \square$

## B  Why we use another network to generate the last-layer bias?

To model model the dependency of the last-layer bias on the activation pattern, we use a neural network, rather than a simpler, linear, and learnable function. The reason is that the bias does not always depend linearly on the activation pattern. Here is an example to illustrate this. There are two outcomes with values $\mathbf{v} = [20, 1]$, four actions with costs $\mathbf{c} = [1.0, 2.1, 2.3, 4.7]$, and the action-outcome transition kernel is

$$P = \begin{bmatrix} 0.211 & 0.789 \\ 0.398 & 0.602 \\ 0.430 & 0.570 \\ 0.684 & 0.316 \end{bmatrix}.$$

Suppose we consider linear contracts, where $\mathbf{f} = \alpha \mathbf{v}, \alpha > 0$. Then the principal's utility function for different contracts is:

$$u^p(\alpha) = \begin{cases} -5\alpha + 5 & 0.2 < \alpha < 0.3 \\ -8.57\alpha + 8.57 & 0.3 < \alpha < 0.4 \\ -9.17\alpha + 9.17 & 0.4 < \alpha < 0.5 \\ -14\alpha + 14 & \alpha > 0.5 \end{cases}.$$

Suppose that we have a 2-dimensional activation pattern, and the linear function converting activation patterns to the bias has parameters $[b_1, b_2]$. Then the bias for each of the four pieces would be $0, b_1, b_2$, and $b_1 + b_2$, respectively. The difference between each piece's bias needs to model the discontinuity at contract parameter $\alpha = 0.3, 0.4, 0.5$, but this is impossible with this linear model. To see this, we first assume that the piece $0.2 < \alpha < 0.3$ has bias 0. Then the differences of biases of the other 3 pieces would need to be 2.5, 2.86, and 5.28, which cannot be achieved with $b_1, b_2$, and $b_1 + b_2$. It can be easily verified that the cases where other pieces have a bias of 0 are similar, demonstrating that a linear bias function cannot express the discontinuity. By contrast, appealing to a second network allows for non-linear dependency on activation, and can handle this problem.

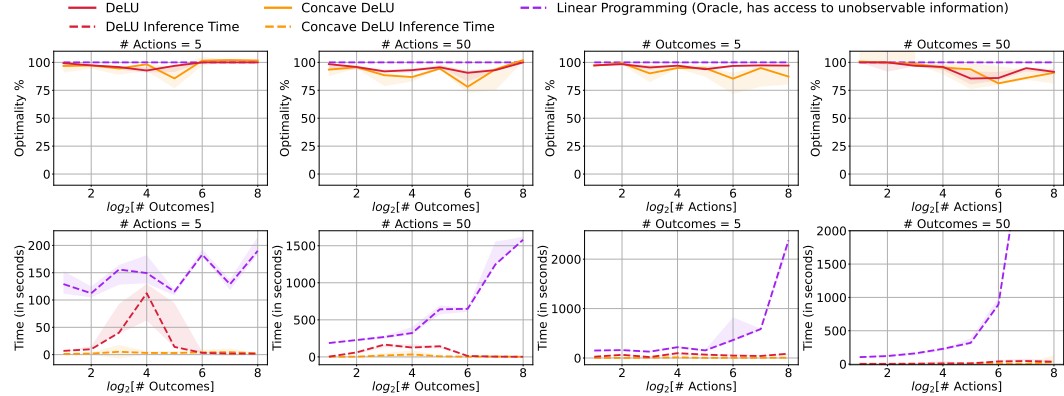

Figure 8: Optimality (normalized principal utility, divided by the result obtained by the direct LP solver `Oracle LP`) and inference time of DeLU, `Concave DeLU`, and `Oracle LP` on increasing problem sizes.

## C   Introducing Concavity into the DeLU Network

From Lemma 4, the principal's utility function $u^p$ is a summation of a concave function and a piecewise constant function. Further, our DeLU network, which is used to approximate the function $u^p$, can be written as

$$\xi(\boldsymbol{f}_i; \theta_\eta, \theta_\zeta) = \eta(\boldsymbol{f}_i; \theta_\eta) + \zeta(r(\boldsymbol{f}_i); \theta_\zeta). \tag{41}$$

In particular, $\zeta(r(\boldsymbol{f}_i); \theta_\zeta)$ is a piecewise constant function, because given an activation pattern $r(\boldsymbol{f}_i)$, $\zeta$ is a constant. However, the first term in Eq. 41, in a general DeLU network, is an arbitrary function. This raises the question as to whether it is useful to further restrict the network architecture, constraining the sub-network $\eta$ to be a concave function. In this section, we discuss how to introduce concavity into $\eta$, and how this restriction affects the performance of DeLU-based contract design.

### C.1   Concave DeLU architecture

We can make $\eta$ a concave function by (1) enforcing its weights (for all but the first layer) to be non-negative, and (2) taking the negation of its output. The first modification will make $\eta$ a convex function because $\eta$ uses ReLU activation, which is a convex and non-decreasing function. When the weights after a ReLU activation are non-negative, $\eta$ becomes a composition of several convex, non-decreasing functions, which is still a convex function. Since the negation of a convex function is a concave function, the second modification will make $\eta$ concave.

Formally, in `Concave DeLU`, the sub-network $\eta$ calculates

$$\boldsymbol{h}^{(L)}(\boldsymbol{x}) = |\boldsymbol{W}^{(L)}|\boldsymbol{R}^{(L-1)}(\boldsymbol{x}) \left( \cdots \left( |\boldsymbol{W}^{(2)}|\boldsymbol{R}^{(1)}(\boldsymbol{x}) \left( \boldsymbol{W}^{(1)}\boldsymbol{x} + \boldsymbol{b}^{(1)} \right) + \boldsymbol{b}^{(2)} \right) \cdots \right) + \boldsymbol{b}^{(L)},$$
$$\eta(\boldsymbol{x}) = -|\boldsymbol{W}^{(L+1)}|\boldsymbol{R}^{(L)}\boldsymbol{h}^{(L)}(\boldsymbol{x}), \tag{42}$$

where $\boldsymbol{R}^{(k)}$ is a diagonal matrix with diagonal elements equal to the layer activation pattern $\boldsymbol{r}^{(k)}$. The other components, as well as the training process, of `Concave DeLU` are the same as a DeLU network. In the next sub-section, we evaluate the performance of `Concave DeLU`.

### C.2   Experiments on Concave DeLU networks

In Fig. 8, we compare `Concave DeLU` against DeLU and the direct LP solver (`Oracle LP`). For this, we fix DeLU and `Concave DeLU` to have the same size, and both use the LP inference algorithm. We test different problem sizes. In the first and second column, we set the number of actions $n$ to 5 and 50, respectively, and increase the number of outcomes $m$ from $2^1$ to $2^8$. In the third and fourth column, we set the number of outcomes $m$ to 5 and 50, respectively, and increase the number of actions $n$ from $2^1$ to $2^8$. For each problem size, the same 12 combinations of $\alpha_p$ and $\beta_p$ are tested. The median

performance as well as the first and third quartile (shaped area) of these 12 combinations are shown. The first row compares optimality, while the second row compares inference time efficiency. We again report normalized principal utilities when it comes to optimality.

A surprising result is that for most problem sizes, DeLU achieves better optimality than `Concave DeLU`. Although the function class represented by `Concave DeLU` is a better fit with $u^p$, it seems that the larger function class of DeLU aids optimization and leads to better performance. At the same time, it is somewhat surprising that `Concave DeLU` can reduce inference time substantially. Unlike DeLU, the inference time of `Concave DeLU` remains relatively stable when the problem size increases. We conjecture that this behavior can be attributed to the non-negativity constraint on network weights, which reduces the number of valid activation patterns and speeds up LP inference.

## D Experimental Setups

### D.1 Network architecture and training

We use a simple architecture for the DeLU network. In all experiments, the sub-network $\eta$ has one hidden layer with 32 hidden units and ReLU activations. We deliberately restrict the size of this sub-network to limit the number of valid activation patterns and speedup LP-based inference. However, this restriction may reduce the representational capacity of DeLU networks: it is in contrast to the common practice of overparameterization, which has contributed to the success of deep learning. To alleviate this concern, we employ a relatively larger network for the bias network $\zeta$. In our experiments, $\zeta$ has a hidden layer with 512 (Tanh-activated) neurons.

The two sub-networks $\eta$ and $\zeta$ are trained in an end-to-end manner by the MSE loss (Eq. 5). The optimization is conducted using RMSprop with a learning rate of $1 \times 10^{-3}$, $\alpha$ of 0.99, and with no momentum or weight decay. For the DeLU and the baseline ReLU networks, training samples are randomly shuffled in each of the training epochs.

### D.2 Infrastructure

Across all experiments, DeLU, and the baseline ReLU networks are trained on a NVIDIA A100 GPU. Gradient-based inference is also parallelized on the A100 GPU. The direct LP solver (`Oracle LP`) and the LP-based inference algorithm are based on the linear programming toolkit PuLP [47], and we parallelize five LP solvers on CPUs.

## E More Experiments

In this section, we carry out experiments to study (1) using random samples to initialize gradient-based inference (Appx. E.1); and (2) the influence of cost correlation on the optimality of DeLU learners (Appx. E.2).

### E.1 Gradient-based inference initialized with random samples

In Sec. 4.2.2, we introduce a gradient-based inference algorithm, and Alg. 1 gives the matrix-form expression of its parallel implementation. There is a choice in using Alg. 1, as to whether the input ("probe points") $X^{(0)} \in \mathbb{R}^{K \times m}$ is taken from the training set or a different, random sample set. In this section, we report the results of experiments to empirically compare these two setups.

We start from the same trained DeLU networks on small ($n = 5, m = 16$), middle ($n = 32, m = 50$), and large ($n = 50, m = 128$) problem sizes, where $n$ is the number of actions and $m$ is the number of outcomes. For each problem size, we consider 12 different combinations of $(\alpha_p, \beta_p)$ as in other experiments and report the mean and variance of the performance. We run Alg. 1 with two different inputs $X^{(0)}$: `Training Set` uses $50K$ training samples while `Random Set` uses $50K$ randomly generated contracts. In Table. 1, we present the normalized principal utility (divided by the result obtained by `Oracle LP`) of these two setups.

We can observe that the optimality of these two setups are very close, especially when the problem size is large. We thus recommend running Alg. 1 initialized with the training set to reduce the possible time and memory overhead of generating a new random sample set.

Table 1: Optimality (normalized principal utility %) of two setups of gradient-based inference: using the training set (`Training Set`) or a random sample set (`Random Set`) as input $\boldsymbol{X}^{(0)}$ of Alg. 1. Mean and variance over 12 different combinations of $(\alpha_p, \beta_p)$ are shown. In this table, the problem size is defined by $(n, m)$, where $n$ is the number of actions and $m$ is the number of outcomes.

| Problem size $(n, m)$ | `Training Set` | `Random Set` |
|---|---|---|
| (5, 16) | $95.29 \pm 0.20$ | $95.33 \pm 0.19$ |
| (32, 50) | $88.43 \pm 0.97$ | $88.46 \pm 0.98$ |
| (50, 128) | $94.28 \pm 0.02$ | $94.28 \pm 0.02$ |

Table 2: Optimality (normalized principal utilities %) of DeLU networks under different combinations of $(\alpha_p, \beta_p)$. Mean and variance over 32 problem sizes are shown.

| | $\alpha_p = 0.5$ | $\alpha_p = 0.7$ | $\alpha_p = 0.9$ |
|---|---|---|---|
| $\beta_p = 0.0$ | $88.79 \pm 12.77$ | $89.67 \pm 12.57$ | $87.20 \pm 11.97$ |
| $\beta_p = 0.3$ | $93.94 \pm 11.16$ | $93.28 \pm 11.48$ | $87.94 \pm 13.45$ |
| $\beta_p = 0.6$ | $95.22 \pm 7.93$ | $95.08 \pm 8.78$ | $91.97 \pm 10.32$ |
| $\beta_p = 0.9$ | $97.23 \pm 6.62$ | $90.69 \pm 18.95$ | $96.43 \pm 7.45$ |

## E.2 Influence of correlated costs

Across our experiments, we test 12 different combinations of $\alpha_p$ and $\beta_p$, where $(\alpha_p, \beta_p) \in \{0.5, 0.7, 0.9\} \times \{0, 0.3, 0.6, 0.9\}$. Recall that a greater $\beta_p$ value means we put more weights on the independent cost, and a greater $\alpha_p$ value indicates that the correlated cost is more close to the expected value of the action. It is interesting to investigate the influence of these two parameters on the performance of DeLU contract designers.

In Table 2, we show the optimality of DeLU learners (using the LP inference algorithm) under different $\alpha_p$ and $\beta_p$ values. For each $(\alpha_p, \beta_p)$ combination, we test 24 different problem sizes: $(m, n) \in \{25, 50, 100\} \times \{2, 4, 8, 16, 32, 64, 128, 256\}$. We give the median and standard deviation of these 24 instances. We observe that $\beta_p$ exerts influence on the performance of DeLU networks: optimality of the learned contracts generally increases for greater values of $\beta_p$, whatever the value of $\alpha_p$, and we see that DeLU seems better suited to handle problems in which the costs of actions are relatively independent of the expected values of actions. This claim is also supported by the results regarding $\alpha_p$, where the optimality under $\alpha_p = 0.5$ is typically better than those under other $\alpha_p$ values for most $\beta_p$ values. It will be interesting in future work to further study this phenomenon, and to see whether suitable modifications can be made to the DeLU architecture to further improve optimality in regimes with higher cost correlation.

