# OpenReview forum: "Deep Contract Design via Discontinuous Networks"
_NeurIPS.cc/2023/Conference — NeurIPS 2023 poster_

### Official Review · Reviewer_GXTm · 2023-07-03

**Soundness:** 3 good
**Presentation:** 3 good
**Contribution:** 3 good
**Rating:** 6
**Confidence:** 4

**Summary:**

This paper studies the problem of contract design, focusing on a scenario where a single principal aims to design the reward structure for different outcomes, followed by a self-interest agent. The principal's utility function, which relies on the agent's best response strategy, is a piecewise affine funciton with points of discontinuity at the boundaries. Based on that, the authors introduce a novel approach by using a discontinuous ReLU (DeLU) network to model the principal's utility function.

To infer the optimal contract using a trained DeLU network, the authors propose two inference methods: a Linear Programming (LP)-based approach and a gradient-based approach.

The authors provide empirical evidence through various experiments to showcase the optimality, sample complexity, and time efficiency of their DeLU network in comparison to the conventional ReLU network, as well as the effectiveness of their proposed inference techniques.


**Strengths:**

1. The idea of representing the principal's utility function using a neural network is novel and insightful. It paves the way for a new research direction in contract design that harnesses the power of neural networks.
2. The motivation behind the DeLU network is well-founded. The authors thoroughly analyze the piecewise affine linear nature of the principal's utility function, including its discontinuity at boundary points. Subsequently, they construct the DeLU network, which aligns with and preserves these crucial properties.
3. The paper exhibits excellent organization throughout, seamlessly integrating theoretical analysis, methodology descriptions, and the experimental section. The logical flow makes it easy to follow and comprehend the content.


**Weaknesses:**

From my view, one significant weakness of the paper is the lack of motivation for utilizing deep learning in the context of optimal contract design.  Why is deep learning a relevant and beneficial approach in this domain? To enhance the paper, it would be essential to address this gap and explain the motivation behind incorporating deep learning techniques for contract design.

For instance, researchers have explored deep learning for optimal auction design because the theoretical research on optimal auctions has encountered bottlenecks. Similarly, it is crucial to identify the motivation for leveraging deep learning in contract design.


**Questions:**

1. As I mentioned in 'Weakness', what is the significance of using deep learning in optimal contract design? why do we care about this?
2. Why model the utility function and infer the optimal contract instead of directly modeling the contract and optimizing it based on the utility function?


**Limitations:**

The authors have discussed the limitations in Section 6.

---

> ### Author Rebuttal · Authors · 2023-08-08
>
> > (Q1) _`What is the significance of using deep learning in optimal contract design? Why do we care about this?`_
>
> The reason, which is similar to the case of auction design, is that previous theoretical and empirical work has encountered bottlenecks.
>
> A particularly interesting angle is the computational one, given that many contracting settings exhibit combinatorial structure, in which case vanilla LP-based approaches fail to be efficient. Indeed, a recent literature in TCS has embarked on designing worst-case poly-time (approximation) algorithms for different combinatorial domains (e.g., [1, 2, 3, 4]). Our approach provides a scalable, general purpose, and beyond-worst case approach for computing (near-)optimal contracts in such settings.
>
> Considering learning theory, whereas there are exponential worst-case sample complexity bounds coming from recent work [5] (exponential  in the number of samples in order to learn an approximately optimal contract, and thus  severe difficulty when the number of outcomes is large), we show empirically that we can get good results with a relatively small number of samples. We see this as a strength of the proposed framework, which is simple to implement, versatile, and general purpose. Getting a better theoretical understanding beyond the worst-case, and in the offline setting of the present paper, is an open problem.
>
> In summary, _no previous method can scale well to the general case of contract design_, motivating us to explore deep learning methods. Moreover, _we expect deep contact design to become more important in the near future_, considering both the digital economy and likely near-term application of AI for automating economic decision making (e.g., contracting with an LLM to plan a vacation, with contracts on outcomes, and the application of smart contracts in DeFi, for example for trading and loans). It seems likely that technological approaches can find application to the design contracts in these kinds of settings [6, 7].
>
> [1] M. Babaioff, M. Feldman, N. Nisan. Combinatorial Agency. EC'06
>
> [2] P. Duetting, T. Roughgarden, I. Talgam-Cohen. The Complexity of Contracts. SODA'20
>
> [3] P. Duetting, T. Ezra, T. Kesselheim, M. Feldman. Combinatorial Contracts. FOCS'21
>
> [4] P. Duetting, T. Ezra, T. Kesselheim, M. Feldman. Multi-Agent Contracts. STOC'23
>
> [5] Zhu, B., Bates, S., Yang, Z., Wang, Y., Jiao, J. and Jordan, M.I., 2023. The sample complexity of online contract design. ACM EC 2023
>
> [6] Horton, J.J., 2023. Large language models as simulated economic agents: What can we learn from homo silicus? (No. w31122). National Bureau of Economic Research.
>
> [7] Park, J.S., O'Brien, J.C., Cai, C.J., Morris, M.R., Liang, P. and Bernstein, M.S., 2023. Generative agents: Interactive simulacra of human behavior. arXiv:2304.03442
>
> &nbsp;
> > (Q2) _`Why model the utility function and infer the optimal contract instead of directly modeling the contract and optimizing it based on the utility function?`_
>
> (1) A first reason is to handle a black box model of an agent, rather than explicitly modeling its actions. A direct optimization approach would need to be able to ``push gradients through" the agent's decision function, along with the effect of this decision (i.e., action) on the world (i.e., outcomes).  For example, if using an approach similar to RochetNet/MenuNet, then the network would model a set of choices for an agent, with each choice corresponding to an action. As a result, the agent's actions would need to be explicitly modeled in the network architecture, which will get infeasible, for example with an exponential number of actions. In our framework, the agent's actions are implicit, and it is only the "contract to utility" function that is modeled by the network.
>
> (2) Direct optimization has been used for auction design. However, the difference there is that auction design is done against a _distribution_ of agent types. _This makes the revenue function in auction design continuous_, with a continuous measure of the type distribution "peeling off" and choosing a different menu choice as the specification of the choices change. Here, we optimize against a single agent type, which makes the principal's utility function discontinuous. For this reason, conventional gradient-based optimization would not be expected to converge to even locally-optimal contracts because the gradients at the boundary are not well defined.

---

### Official Review · Reviewer_GYpK · 2023-07-06

**Soundness:** 4 excellent
**Presentation:** 4 excellent
**Contribution:** 3 good
**Rating:** 7
**Confidence:** 4

**Summary:**

The authors focus on the problem of contract design. In this problem, there is an agent which can take some costly action, each action resulting in a distribution over possible outcomes. A principle gets utility based on outcomes, and incentivizes the agent to take actions which will benefit them by transferring them a payment that depends on the outcome. The agent chooses the action that maximizes their expected received payment minus cost; the principal gets a utility equal to the utility of the outcome minus the required payment.

In mechanism design for selling goods (e.g. auctions) there is a recent thread of work on “differentiable economics” for learning good mechanisms. The authors to some extent work within this broad area, although the techniques they represent depart from prior work in interesting ways. In particular, rather than using a differentiable neural network to represent the mechanism or the agent’s utility as a function of agent types, they instead represent the principal’s utility as a function of the mechanism (contract) itself. The authors present a new network architecture (DeLU) well-suited to learning to represent principal utilities (which are always piecewise affine, though unlike ReLU network not necessarily continuous). By training these networks in a supervised manner, the authors approximate the utility of a given contract. They then present various techniques to optimize over network inputs to find a high-utility contract, and show that these approaches do in fact give good-performing contracts.

**Strengths:**

The problem of contract design is relevant and interesting, and the authors tackle it in a novel way. The new network architecture and new techniques for optimizing over its input are interesting technical contributions. The experiments are carefully done and support the empirical claims made.

**Weaknesses:**

A major concern I have is that essentially, the authors take many examples of contracts, use them to supervised-learn an approximate utility function, and then optimize on that utility function. Does this reliably work much better than just taking the best contract observed from the training data, skipping this intermediate step? This general problem also shows up in other mechanism design problems where utility functions are learned via supervised learning -- does learning the utility function and using it significantly improve downstream performance compared to just using the fixed training dataset as a "pointwise" utility function?

I don't think this is a trivial issue and if unaddressed it is hard to tell if the paper's contribution serves a purpose or not. This is the main reason for my low score and if it can be addressed and no other problems emerge during the discussion phase, I would significantly raise my score.

After author response: this issue is satisfactorily addressed.

**Questions:**

Figure 1, although a valuable and informative plot, is unpleasant looking. Now that you are not up against NeurIPS deadline it might make sense to take the time to do it and make it better looking (e.g. change the strange camera perspective, make axes not cut off, etc.)

How would this approach differ from directly doing gradient-based optimization in the space of contracts, similar to RochetNet/MenuNet for auctions?

It is possible to embed ReLU networks in a mixed-integer program and globally optimize over their inputs. This approach, along with some preprocessing and pruning, has been shown to work quite well even at relatively large scales (e.g. Tjeng et al., https://arxiv.org/abs/1711.07356), especially since good MIP solvers are so fast these days. Could this technique be used effectively to globally optimize over DeLU networks too?


**Limitations:**

The authors adequately discuss limitations of their work, except for the issue mentioned in "Weaknesses". I think better discussion of the size of problem instances in which their approach can be used would be warranted.

---

> ### Author Rebuttal · Authors · 2023-08-08
>
> > (Major concern) _`Does learning the utility function and using it significantly improve downstream performance compared to just using the fixed training dataset as a "pointwise" utility function?`_
>
> Thanks for this question. We present the results from a new set of experiments to demonstrate that learning and then maximizing utility functions finds significantly better contracts than those in the training samples, especially when the number of outcomes is large.
>
> __Experiments.__ We first compare the utilities of the best contracts in the training datasets and the contracts found by DeLU.
>
> In Table 2, we fix the number of actions $n$ to 50 and increase the number of outcomes $m$ from $2^5$ to $2^8$. As in other experiments, for each problem size, 12 combinations of $\alpha_p$ and $\beta_p$ are tested, and we report the median optimality (normalized principal utility) and standard deviation.
>
> __Table 2. Optimality of the best training sample and DeLU contract.__  The number of actions $n$ is fixed to 50.
> |# Outcomes|32|64|128|256|
> |-|-|-|-|-|
> |Best Training Samples|85.83±2.05%|78.72±2.54%|72.04±2.24%|65.41±2.07%|
> |DeLU|92.62±3.81%|97.22±3.71%|88.64±6.93%|94.14±6.27%|
>
> When the number of outcome is 256, the best training sample has an optimality of 65.49%, significantly lower than 94.13% achieved by DeLU. Moreover, the optimality of the best training sample decreases as the number of outcomes increases, while our method scales well with the number of outcomes. DeLU also performs substantially better than the best training sample when we increase the number of actions (Table 3). In Figure 1 of the PDF attached in the general response, we observe similar results of these two methods in more problem sizes.
>
> __Table 3. Optimality of the best training sample and DeLU contract.__ The number of outcomes $m$ is fixed to 50.
> |# Actions|32|64|128|256|
> |-|-|-|-|-|
> |Best Training Samples|73.34±3.41%|77.37±3.20%|73.16±3.19%|70.68±2.29%|
> |DeLU|96.44±5.68%|89.84±9.07%|95.03±5.31%|94.30±3.07%|
>
> The advantage of supervised learning is more significant in very large scale problems. In Table 4, we can see that in problems with 1M outcomes, the performance of DeLU is about 2x  the performance of the best training sample.
>
> __Table 4. Utility of DeLU contracts / utility of the best contract in the training set in large-scale problems.__
> |(# Outcomes, # Actions)|(1K,1K)|(10K,10K)|(50K,2K)|(100K,1K)|(1M,100)|
> |-|-|-|-|-|-|
> |DeLU Result / Best Training Sample|166.49%|190.54%|194.72%|197.64%|199.18%|
>
> __Why?__ As Lemma 3 shows, the  optimal contract is on the boundary of a linear piece (at an optimal solution, the agent's utility of taking the  actions corresponding to adjacent contracts should be the same). This sub-space in which these optimal contracts reside is of a lower dimension. As the dimension of the contract space grows, the probability of obtaining a random sample in this lower-dimensional sub-space gets closer to 0.
>
> &nbsp;
> > (Q2) _`How would this approach differ from directly doing gradient-based optimization in the space of contracts, similar to RochetNet/MenuNet for auctions?`_
>
> A first difference is that auction design is done against a _distribution_ of agent types, which makes the revenue function in auction design  continuous (with a continuous measure of the type distribution "peeling off" and choosing a different menu choice as the specification of the choices change). Here, we optimize against a single agent type, which makes the principal's utility function discontinuous. For this reason, direct gradient-based optimization would not be expected to converge to even locally-optimal contracts because the gradients at the boundary are not well defined.
>
> A second difference in applying RochetNet/MenuNet is that the choice in the menu would correspond to an action of the agent (it is not choosing from a menu of contracts, but choosing from a menu of actions given a single contract). For this, the actions would need to be explicitly modeled in the network architecture, and this will become infeasible, for example in the case of an number of actions that increases exponentially in some natural domain parameters.  In our framework, the agent's actions are implicit, and it is only the "contract to utility" function that is modeled by the network.
>
> &nbsp;
> > (Q3) _`Could this technique (embed ReLU networks in a mixed-integer program and globally optimize over their inputs) be used effectively to globally optimize over DeLU networks too?`_
>
> Thanks for this inspiring question. Globally optimizing a DeLU network involves the bias term for the last layer. This piecewise bias is generated by a Tanh-activated multi-layer network, which is challenging to be embedded in a mixed-integer program (MIP). But if we use ReLU as the activation function in this bias network, then it would be in principle possible to test a global MIP method for inference (our intuition is this would scale less well than the current, piece-oriented inference method).
>
> &nbsp;
> > (Limitations) _`I think better discussion of the size of problem instances in which their approach can be used would be warranted.`_
>
> In Table 4, we test our method empirically on problems with up to 1M outcomes. As the standard LP algorithm becomes intractable, we compare DeLU results to the best training samples, and DeLU shows substantially superior outcomes.
>
> Given sufficient RAM and GPU RAM (we now use 80G RAM and an A100 GPU with 40G GPU RAM, allowing at most 10K training samples for 1M outcome problems), we anticipate applicability of our method to even larger problems. This is because a problem size depends on actions and outcomes, both of which DeLU well handles: (1) actions impact linear pieces count, and DeLU represents an exponentially large number of linear pieces (e.g., $2^{128}$ pieces by a single ReLU layer with 128 units); and (2) outcomes affect input dimension, and neural networks are known to be good at handling high-dimensional inputs.

---

> > ### Comment · Reviewer_GYpK · 2023-08-10
> > **Thanks for response**
> >
> > Your new experiments have successfully addressed my major concern. I don't think the full description of the experiments needs to take up space in the main paper, but some mention in the text + full description in the appendix seems important to me.
> >
> > Thanks for your thoughtful responses to the other issues.
> >
> > I will raise my score to "7" as soon as I can figure out where the edit button is on OpenReview.

---

> > > ### Author Response · Authors · 2023-08-10
> > > **Thanks for your response! We will incorporate the new experiments into the main paper.**
> > >
> > > Thank you for your prompt reply! We are pleased that our response addressed your concern.
> > >
> > > We concur that it is very important to include the new experiments in the main paper. We intend to:
> > >
> > > - Introduce _`Best Training Sample`_ as a new baseline (around Line 288).
> > > - Update Fig. 4 of the main paper to include plots of this baseline.
> > > - Provide a (concise) description and analysis of the results (around Line 305).
> > > - Describe the setup and results in detail in Appendix C.
> > >
> > > We hope these revisions can assist readers in better telling the contribution of our work.

---

### Official Review · Reviewer_1ATX · 2023-07-09

**Soundness:** 4 excellent
**Presentation:** 4 excellent
**Contribution:** 4 excellent
**Rating:** 8
**Confidence:** 4

**Summary:**

This paper considers an offline learning problem of optimal contract through neural network. The authors propose a novel neural network architecture, called Discontinuous ReLU (DeLU) network, which models a piecewise affine function with discontinuous boundaries --- a representation that captures the principal (contract designer)'s utility function with respect to the different contracts. With the neural network to model the principal's utility, the paper showcases two methods to determine the optimal contract, linear programming or the gradient-based interior-point method. The gradient based method is shown to be more efficient in the experiments.

**Strengths:**

1. The paper is well written and easy to follow. The authors provide a good introduction to the contract design problem and the related literature. The paper is also well organized, with a clear description of the proposed method and the experiments. The figures in this paper are very well designed for readers to intuitively understand the idea.
2. The paper proposes a novel neural network architecture for training a piecewise affine function with discontinuous boundaries, as well as the gradient based optimization for inference. This architecture is very interesting and useful for many other applications in multi-agent learning. The authors also provide a good explanation of the architecture and the intuition behind it.
3. The paper provides extensive empirical experiments on simulated data with comparison of different network architecture, inference methods.


**Weaknesses:**

1. One major concern on the proposed method of this paper is the fact that it attempts to approximate the optimal contract simply as the argmax to the approximated principal’s utility. This does not seem to be a reasonable choice, because there may be a constant gap between the expected agent response and the actual agent response. In particular, the argmax contract at some boundary of the piecewise affine function in the approximated principal’s utility (as pointed out in Lemma 3) is not robustified from the small inaccuracies at the boundary --- at least I do not see how the proposed method can ensure an accurate boundary can be learnt. And this necessarily leads to a constant drop in the testing against the actual agent in a large class of problem instances. If this understanding is indeed correct, it is actually surprising that the proposed method can still achieve a good performance in the experiments. The authors should provide some explanation on this issue; I suspect this is due to the special structure in the synthesized data.



2. The paper in general lacks theoretical analysis.  It would be interesting to see some theoretical analysis on the generalization error/sample complexity of the proposed network architecture. The authors could also provide some theoretical analysis on the convergence of the gradient based optimization method, as well as the convergence of the linear programming method.

3. I also expect the paper to include some discussion on the novelty of neural network architecture. I wonder if similar attempts have been made to design neural architecture for special function structures in the hypothesis class.

> Both the first and second concern are resolved given the additional experiments and details provided by the authors in the rebuttal.


**Questions:**

Please answer to my concern in the first point of weakness section.

---

> ### Author Rebuttal · Authors · 2023-08-09
>
> >(Major concern) `One major concern... is that it attempts to approximate the optimal contract as the argmax to the approximated principal’s utility. In particular, the argmax contract at some boundary... is not robustified from the inaccuracies at the boundary.`
>
> Thanks for this point, which motivates us to consider additional analyses and experiments to support the following three arguments, which we hope can address your concern.
>
> ___The current alignment degree between DeLU and real boundaries can support the good performance of DeLU.___ Although we didn't explicitly discuss this in the paper, we believe the reason for this is that  MSE loss is sensitive to misalignment between real and DeLU boundaries. In particular, the jump of the utility function at boundary points can be  arbitrarily large, and thus, a slight misalignment between DeLU and real boundaries can lead to a large increase in the MSE loss.
>
> We conduct additional experiments to test this viewpoint. For each contract design problem, we randomly sample a large number (50K) of contracts, and check whether they are simultaneously on the real and the DeLU boundary. Specifically, we randomly sample 10K directions for each contract and assess linearity of the real and DeLU utility function in each direction. If the function is non-linear in some (>20%) directions, we mark the contract as on a boundary. In Table 1, we report the percentage of overlapped boundaries points (# samples on both real and DeLU boundaries / # samples on real boundaries). Here we fix the number of outcomes to 5, and increase the number of actions from $2^2$ to $2^8$. For each problem size, we present the median and s.d. of 12 different $(\alpha_p,\beta_p)$ combinations. We observe that DeLU achieves a good degree of boundary alignment.
>
> Table 1. DeLU learns boundaries reasonably aligned with real boundaries, and the performance of DeLU is related to the boundary alignment degree.
> |# Actions|4|8|16|32|64|128|256|
> |-|-|-|-|-|-|-|-|
> |Overlapped boundary points (%)|99.71±12.40|83.09±17.32|74.54±15.20|91.36±10.23|76.44±17.36|98.86±4.41|80.57±12.65|
> |DeLU optimality (%)|95.81|88.54|91.78|91.13|89.30|93.02|88.87|
>
> Shown in the second row is the optimality (normalized principal utility) of DeLU contracts. It can be observed that the boundary alignment degree is related to DeLU (argmax) performance.
>
> ___Our method can be extended to support "sub-"argmax, which slightly improves its performance.___ Following the reviewer's comments, going beyond argmax is also possible with our gradient-based inference. When annealing the coefficient of the barrier function $1/t^{(k)}$, we can check whether the actual principal utility increases for each $t$ value. If the utility decreases, we know that we encounter inaccurate boundaries and can stop the inference to seek more robustness. Tested on 8 different problem sizes, we find that "early stop" can increase the optimality by 3.96±0.20% (avg±var). Thanks to the reviewer for inspiring this mechanism, which will be included in our codebase. But we also note that this performance improvement is not very large, in line with DeLU boundaries aligning reasonably with real boundaries.
>
> ___DeLU recovers optimal contracts on a range of problems.___ The reviewer asks whether the good performance is due to the special structure in synthesized data. We take DeLU to a nonlinear contract design problem with some real-world economic intuition, presented in STOC 2022 and EC 2019 tutorial ([1], Page 47). DeLU achieves an optimality of 97.83% (2.89/2.95). Furthermore, we consider a wide range of environments in our experiments including correlations of different kinds. The results in Figs 4-6 are presented for 12 different correlation structures. Table 2 (Appendix) gives a breakdown for these structures and shows robust performance.
>
> [1] Duetting, P. and Talgam-Cohen, I., Contract Theory: A New Frontier for AGT
>
> &nbsp;
> > `The novelty of neural network architecture.`
>
> A longstanding challenge in the deep learning community has been approximating discontinuous functions using neural networks. While the Universal Approximation Theorem only guarantees the approximation of continuous functions, many crucial problems involve discontinuity, such as astronomy (e.g., solar flare imaging [2]) and mathematics (e.g., uncertainty quantification [3]).
>
> However, establishing a discontinuous network is not an easy task. Discontinuities were considered as early as in the 1950s when neural networks were first proposed [4] by using step activation functions. Following this work, most models introduced discontinuity through the use of different discontinuous activation functions. However, optimizing these models is typically more challenging than with continuous activation functions [3], hindering the application of discontinuous networks. To the best of our knowledge, this paper presents the first discontinuous network architecture with continuous activation functions and stable optimization performance.
>
> For economics, we expect that our exploration of discontinuous networks can draw attention to problems involving discontinuity, especially in utility functions. For example, this kind of discontinuity also arises in mechanism design when agents, in effect, ``chose'' from a menu of options. For this reason, we are hopeful that further research on discontinuous network architectures and optimization methods can contribute to advancing AI progress in computational economics.
>
> [2] Massa, P., Garbarino, S. and Benvenuto, F., 2022. Approximation of discontinuous inverse operators with neural networks. Inverse Problems, 38(10), p.105001
>
> [3] Della Santa, F. and Pieraccini, S., 2023. Discontinuous neural networks and discontinuity learning. Journal of Computational and Applied Mathematics, 419, p.114678
>
> [4] Rosenblatt, F., 1958. The perceptron: a probabilistic model for information storage and organization in the brain. Psychological review, 65(6), p.386

---

> > ### Comment · Reviewer_1ATX · 2023-08-14
> >
> > Thanks for the detailed response. The additional experiments on the alignment of decision boundary boosts my confidence in the techniques proposed by this work; to some degree, I find it magical and definitely worth follow-up studies from both theoretical and empirical sides. However, I still have concerns on these new experiment results. Therefore, I am only willing to increase my score if the author could provide some important details in the new experiments: (1) how many samples are used to train the DeLU networks in each instances?   (2) Why is this approach reasonable, "If the function is non-linear in some (>20%) directions, we mark the contract as on a boundary."? Can you formally describe your algorithm to check whether a sampled contract is on the boundary?
> >
> > Below are some of my additional comments:
> >
> > - The high variance of "Overlapped boundary points (%)" seems to suggests that there are instances where the decision boundary alignment is bad, and I still suspect there might be adversarial instances where boundary alignment is arbitrarily bad (e.g., sharp drops at every boundary). I also wonder whether these instances where alignment is bad coincides with the instances where "sub-"argmax substantially improves the performance. These findings should help us understand the exact mechanisms undergoing when DeLU approximates the agent's decision function.
> >
> > - I suggest the authors to include (and highlight) these new studies on boundary alignment in the next version of the paper. The current lemmas of Section 3 are well-known in contract design theory and can be significantly enhanced to motivate the study of decision boundary alignment. I think the boundary alignment problem is the key learning challenge in strategic setting, and the paper should deserve higher score if the narrative were to be centered around the "surprising effectiveness of boundary alignment by DeLU networks".
> >
> > - Based on my understanding, the proposed techniques seems to be generalizable to problem setups beyond contract design (e.g., Stackelberg games, security games). It is unclear why the authors choose to focus on the contract design problem (or perhaps it is important that the contract space is unbounded?). I wonder if the authors have any comment on this.

---

> > > ### Author Response · Authors · 2023-08-16
> > > **Thanks for your valuable questions and comments! (Part 1)**
> > >
> > > > (Detail 1) `# Training samples.` All the instances here are trained with 50K samples.
> > >
> > > > (Detail 2) _`Why is this approach reasonable: If the function is non-linear in some (>20%) directions, we mark the contract as on a boundary.`_
> > >
> > > Since the principal's utility function is piecewise linear, the function exhibits linearity in the proximity of an interior point. Conversely, when a point lies on a boundary, there is a jump in utilities within its proximity, rendering it unable to pass a linearity test in some directions.
> > >
> > > Please note that we use exactly the same approach to check for both true and DeLU boundaries.
> > >
> > > &emsp;
> > > > (Detail 3) `The algorithm to check whether a contract is on a true/DeLU boundary` is in lines starting with a square symbol ($\blacksquare$).
> > >
> > > __Algorithm__ Boundary alignment degree calculation
> > >
> > > __Input:__ True principal utility function $u$ and its DeLU approximation $\tilde{u}$
> > > * $n_{true}=0$, $n_{ovlp}=0$&emsp;`Count # true and overlapped boundary points`
> > > * __for__ $k=1,\cdots,K$ __do__:&emsp;`For K random contracts`
> > >   * $\mathbf{f}^k\leftarrow$ a uniform random contract
> > >   * $n_{\text{true-nl}}=0$, $n_{\text{DeLU-nl}}=0$&emsp;`Count # directions in which the true and DeLU function is non-linear`
> > >   * __for__ $n=1,\cdots,N$ __do__:&emsp;`for N random directions`
> > >     * $\mathbf{d}^n\leftarrow$ a uniform random sample in $\mathbb{R}^{m}$
> > >     * $\mathbf{d}^n\leftarrow\delta\frac{\mathbf{d}^n}{||\mathbf{d}^n||}$&emsp;`Normalize`
> > >     * __if__ $u(\mathbf{f}^k)+u(\mathbf{f}^k+2\mathbf{d}^n)\ne 2u(\mathbf{f}^k+\mathbf{d}^n)$:&emsp;`If the true utility function is non-linear`
> > >       * $n_{\text{true-nl}}$+=1
> > >     * __if__ $\tilde u(\mathbf{f}^k)+\tilde u(\mathbf{f}^k+2\mathbf{d}^n)\ne 2\tilde u(\mathbf{f}^k+\mathbf{d}^n)$:&emsp;`If the DeLU utility function is non-linear`
> > >       * $n_{\text{DeLU-nl}}$+=1
> > >   * __if__ $n_{\text{true-nl}}/N>\tau$:&emsp;`If there are many non-linear directions`
> > >     * $n_{true}$+=1
> > >     * __if__ $n_{\text{DeLU-nl}}/N>\tau$:&emsp;`Check whether it is also on the DeLU boundary`
> > >       * $n_{ovlp}$+=1
> > > * return $n_{ovlp}/n_{true}$
> > >
> > > &emsp;
> > > > (Comment 1) _`Whether the instances where alignment is bad coincides with the instances where "sub-"argmax substantially improves the performance.`_
> > >
> > > Thanks for asking about this. In fact, we do generally observe that the performance improvement from the "sub-"argmax method is related to the boundary alignment degree (Table 5).
> > >
> > > __Table 5__. Performance improvement provided by "sub-"argmax, in decreasing order of boundary alignment, on instances with __(a)__ # actions=5, $\beta_p=0.9$, $\alpha_p=0.5$.
> > > |# outcomes|256|8|128|32|4|16|64|
> > > |-|-|-|-|-|-|-|-|
> > > |Boundary alignment degree (%)|93.89|85.99|84.49|84.16|73.80|46.42|33.45|
> > > |Performance improvement (%)|-0.80|+2.08|+0.46|+1.31|+6.27|+2.40|+12.36|
> > >
> > > __(b)__ # actions=5, $\beta_p=0.3$, $\alpha_p=0.9$.
> > > |# outcomes|4|128|32|8|64|256|16|
> > > |-|-|-|-|-|-|-|-|
> > > |Boundary alignment degree (%)|98.14|97.88|93.97|92.90|86.90|84.74|66.59|
> > > |Performance improvement (%)|+0.63|+0.03|+0.17|+0.77|+22.56|+3.63|+3.10|
> > >
> > > &emsp;
> > > > (Comment 2) _`I suggest the authors to include (and highlight) these new studies on boundary alignment in the next version of the paper.`_
> > >
> > > We have found this exchange very useful, and agree that studying boundary alignment is informative. We plan to make the following changes:
> > >
> > > - Motivate and discuss the boundary alignment question after the lemmas in Sec. 3.
> > > - Analyze the influence of MSE loss on boundary alignment in Sec. 4.2.
> > > - Introduce the "sub-"argmax approach in Sec. 4.2.2.
> > > - Add a new experimental subsection studying boundary alignment. Incorporate a detailed description of the boundary check algorithm and experiments on the relationship between alignment degrees, DeLU performance, and improvement provided by "sub"-argmax.

---

> > > > ### Author Response · Authors · 2023-08-16
> > > > **Thanks for your valuable questions and comments! (Part 2)**
> > > >
> > > > &emsp;
> > > > > (Comment 3) _`Why the authors choose to focus on the contract design problem.`_
> > > >
> > > > First of all, contract design is a basic problem in economic theory (including the 2016 Nobel Prize in Economics), and is gaining increased attention in the digital economy. For example, it doesn't seem too unlikely that we will soon have contracts for LLM-style actor [1] or generative models as simulators of human decision making and behavior (and in the future, likely firm behavior) [2].
> > > >
> > > > That said, we also agree with the reviewer that DeLU can be explored in other problems. We're interested to bring attention to economics problems involving discontinuities, as studied here through this network model of discontinuous functions and on the specific problem of contract design. We hope to motivate additional applications, for example to Stackelberg equilibria in other contexts (and we've also studied applications to problems of Bayesian persuasion), and we'll expand on this in the future work discussion. Additional methodological work can also consider network models of piecewise discontinuous but _non-linear_ functions.
> > > >
> > > > [1] Horton, J.J., 2023. Large language models as simulated economic agents: What can we learn from homo silicus? (No. w31122). National Bureau of Economic Research
> > > >
> > > > [2] Park, J.S., O'Brien, J.C., Cai, C.J., Morris, M.R., Liang, P. and Bernstein, M.S., 2023. Generative agents: Interactive simulacra of human behavior. arXiv:2304.03442

---

> > > > > ### Comment · Reviewer_1ATX · 2023-08-16
> > > > >
> > > > > Thanks for the prompt response! They resolve many of my concerns and I am very excited about these discoveries. I have increased my score accordingly.

---

> > > > > > ### Author Response · Authors · 2023-08-16
> > > > > > **Thanks for your feedback!**
> > > > > >
> > > > > > We extend our appreciation to the reviewer for delivering high-quality and insightful reviews and feedback, which have not only illuminated the weaknesses within our paper but also spurred the discovery of new insights, greatly enriching the depth of our work.

---

### Official Review · Reviewer_aB6m · 2023-07-09

**Soundness:** 3 good
**Presentation:** 3 good
**Contribution:** 3 good
**Rating:** 6
**Confidence:** 4

**Summary:**

This paper proposes a deep-learning approach for contract design. The rationale is:
- the principal's utility function should be learned from the data
- the problem of approximating accurately the utility function is non-trivial, and the utility function may be discontinuous --- suggesting the development of better models such as those proposed in the paper

In addition to the model approximation problem, the authors improve the available algorithms for training and inference. Finally, the authors experimentally evaluate their paradigm according to several dimensions.

**Strengths:**

The authors clearly posed the problem and the presentation is clear.

**Weaknesses:**

At the current stage, I have some doubts about the significance of the contributions and I need that the authors help me to understand better that issue.

My feeling is that the machine-learning contribution is not very strong and that the work is primarily a slight variation of deep-learning tools applied to contract design. That is, from a machine-learning perspective, this paper does not provide substantial advancements. This is not necessarily a critical issue for the acceptance of the paper. Many papers just apply machine learning tools to design important applications.  More importantly for me, the advancement in the contract design perspective is not really strong. Yes, the authors are providing better approximations, but it is not clear to me the applicability of the results, e.g., due to the need for thousands of samples for the training and it is unlikely such data are available.

I agree with the authors that the main works done so far focus on online/bandit approaches and that the authors are providing a completely different perspective. While online/bandit approaches are directed at a small number of samples, deep learning makes the reverse and, in my opinion, could require a too large number of samples for a good approximation. Personally, I believe that something in the middle would be useful.

**Questions:**

*Technicalities*
- are the techniques for training and inference provided by the authors important advancements (i.e., non-trivial, non-direct) of the deep learning techniques? if yes, why and how

*Applicability*
- is correct my understanding that several thousands of samples are necessary for training? if yes, do you believe that making thousands of queries to the principal is reasonable (using $10^6$ in the plot does not help and I would suggest the authors use semi-log plots to show what happens with a small number of samples).


**Limitations:**

Not applicable.

---

> ### Author Rebuttal · Authors · 2023-08-09
>
> > (Q1: Technicalities) _`Are the techniques for training and inference provided by the authors important advancements of the deep learning techniques?`_
>
> A longstanding challenge in the deep learning community has been approximating discontinuous functions. While the Universal Approximation Theorem only guarantees the approximation of continuous functions, many crucial problems involve discontinuity, such as astronomy (e.g., solar flare imaging [1]) and mathematics (e.g., uncertainty quantification [2]).
>
> However, establishing a discontinuous network is not an easy task. Discontinuities were considered as early as in the 1950s when neural networks were first proposed [3] by using step activation functions. Following this work, most models introduced discontinuity through the use of different discontinuous activation functions. However, optimizing these models is typically more challenging than with continuous activation functions [2], hindering the application of discontinuous networks. To the best of our knowledge, this paper presents the first discontinuous network architecture with continuous activation functions and stable optimization performance. As other reviewers stated, _"the proposed DeLU network and its concave counterpart are novel and worthy of studying further,"_ and _"the paper proposes a novel neural network architecture ... is very interesting and useful for many other applications in multi-agent learning."_ The proposed inference algorithm also provides a new approach to ReLU maximization. Previous work typically relies on mixed integer linear programming [4], while our method is gradient-based and scales well with the input dimensions and network sizes while matching GPU/TPU architecture.
>
> For economics, we expect that our exploration of discontinuous networks can draw attention to problems involving discontinuity. For example, discontinuity also arises in mechanism design when agents, in effect, ``chose'' from a menu of options. We hope that research on discontinuous network architectures and optimization will contribute to advancing AI progress in computational economics.
>
> [1] Massa, P., Garbarino, S. and Benvenuto, F., 2022. Approximation of discontinuous inverse operators with neural networks. Inverse Problems, 38(10), p.105001
>
> [2] Della Santa, F. and Pieraccini, S., 2023. Discontinuous neural networks and discontinuity learning. Journal of Computational and Applied Mathematics, 419, p.114678
>
> [3] Rosenblatt, F., 1958. The perceptron: a probabilistic model for information storage and organization in the brain. Psychological review, 65(6), p.386
>
> [4] Tjeng, V., Xiao, K.Y. and Tedrake, R., Evaluating Robustness of Neural Networks with Mixed Integer Programming. ICLR 2018
>
> &nbsp;
> > (Q2.1: Applicability) _`Is correct my understanding that several thousands of samples are necessary for training? (... I would suggest the authors use semi-log plots to show what happens with a small number of samples.)`_
>
> The number of training samples depends on the problem size. On relatively small problems (16 outcomes, 5 actions), using 100 samples is enough to achieve an optimality of 92.99%. For large problem sizes (50 actions, 32 outcomes), we need 5K samples to get a satisfactory optimality (94.67%). Fig. 2 in the PDF of the general response presents semi-log plots to show in detail how DeLU performance changes with small numbers of training samples.
>
> Considering learning theory, whereas there are exponential worst-case sample complexity bounds coming from recent work [5] (exponential in the number of samples in order to learn an approximately optimal contract, and thus severe difficulty when the number of outcomes is large), we show empirically that we can get good results with a relatively small number of samples. We see this as a strength of the proposed framework, which is simple to implement, versatile, and general purpose. Getting a better theoretical understanding beyond the worst-case, and in the offline setting of the present paper, is an open problem.
>
> [5] Zhu, B., Bates, S., Yang, Z., Wang, Y., Jiao, J. and Jordan, M.I., 2023. The sample complexity of online contract design. ACM EC 2023
>
> &nbsp;
> > (Q2.2: Applicability) _If yes, do you believe that making thousands of queries to the principal is reasonable?_
>
> We discuss this in different applications to which our method can be applied.
>
> (1) The first application is as a tool for _theoretical economists_ to study contract design problems. In economic theory, one assumes a model of a world environment (e.g., actions, technology, costs, rewards, outcomes), and looks to understand the optimal designs. In this setting, thousands of queries is reasonable because we have a model of the world.
>
> (2) A second application comes from the _digital economy_, where we can expect to gain access to large training sets as we see increasing automation of economic processes (e.g., it doesn't seem too unlikely that we will soon have contracts for LLM-style actors, for example working to plan vacation details for a user with the possibility of contracting on outcomes) [6].
>
> (3) A third application comes from settings where we can expect to have access to a simulator of the agent behavior. In particular, a black-box simulator fits perfectly with our set-up, where we only need access to the induced utility to the principal for different contract designs. These simulations may be for automated agents in the digital economy. Another interesting development is that attention is going these days to _generative models as simulators of human decision making and behavior_ (and in the future, likely firm behavior) [7].
>
> [6] Horton, J.J., 2023. Large language models as simulated economic agents: What can we learn from homo silicus? (No. w31122). National Bureau of Economic Research
>
> [7] Park, J.S., O'Brien, J.C., Cai, C.J., Morris, M.R., Liang, P. and Bernstein, M.S., 2023. Generative agents: Interactive simulacra of human behavior. arXiv:2304.03442

---

> > ### Comment · Reviewer_aB6m · 2023-08-14
> > **Response to the authors**
> >
> > I understand the peculiarity of the problem (learning non-continuous functions) and the technical advancement. I also appreciate the authors' reply about the training data. I agree that similar concerns can be found in other settings in which works on deep learning can be commonly found. However, I keep being skeptical about the actual applicability in real-world applications. I raise my scores accordingly.

---

> > > ### Author Response · Authors · 2023-08-16
> > > **Thanks for your response!**
> > >
> > > Thanks for your feedback! We are delighted that our response can address some of your concerns.
> > >
> > > We have found the review very helpful, and plan to make the following changes to our paper:
> > >
> > > - Motivate and discuss the technical novelties from the perspective of deep learning (around Line 71).
> > >
> > > - Update Figure 6 on Page 9, using semi-log plots to show DeLU performance with a small number of training samples. Also update the accompanying discussion in Sec. 5.
> > >
> > > - Provide a more extensive discussion about real-world applications of the proposed method in a future work section.

---

### Official Review · Reviewer_z4Vn · 2023-07-19

**Soundness:** 4 excellent
**Presentation:** 4 excellent
**Contribution:** 4 excellent
**Rating:** 9
**Confidence:** 3

**Summary:**

This paper introduces an automated optimal contract design method from offline data using deep learning. In this setting, a principle deigns a contract establishing an agreement of payments the principle will undertake for the outcomes arising from the actions of an agent. Given a contract, the agent privately selects an action to maximize its expected utility under an action-outcome transition kernel. The principle, unaware of the agent's actions and the transition kernel, aims to maximize its own expected utility. The authors propose the use of neural network function approximation to learn the principle's utility function from offline samples. Due to piecewise discontinuous nature of the principle's utility function and the continuous nature of existing neural network (eg. ReLU), the authors propose the Discontinuous ReLU (DeLU) network to model the the principle's utility function as a discontinuous piecewise affine function, where each piece represents a particular action taken by the agent. Moreover, the authors present a computationally efficient inference method for contract designing based on interior-point method. Finally, the paper empirically evaluates the validity and the performance of the methods presented here on synthetic data.

**Strengths:**

1) Originality:
Noting that I am not quite familiar with this field, this is the first ever use of neural network learning of utility functions for contract design based on the author's claims. The authors not only combine two existing ideas, but also innovate a new neural network architecture as the existing models are not capable of capturing the critical discontinuities in the utility function. Moreover, they propose an efficient implementation of the inference computations by utilizing highly parallelized existing deep learning frameworks. They also propose an alternative concave neural network architecture.

2) Quality
The claims made here were supported by sound theoretical and empirical evidence.

3) Clarity
Even though I am out of the field, it was very easy to follow the paper. The problem and its motivation are explained very clearly. The theory and the experimental results are stated neatly and discussed properly. It was a pleasure to read this paper.

4) Significance
Again, since this is outside my expertise, it is less obvious to me why this problem should be studied. The examples provided by the authors as well as a quick Google search were helpful to clarify this. Aside from the real-world significance, the proposed DeLU network and its concave counterpart are novel and worthy of studying further.



**Weaknesses:**

I don't have much to say here.

**Questions:**

I have some question to the authors out of curiosity.
1) How easy is it to extend your method to contract with mixed payment-penalty structure (ie. negative values for f)?
2) Is there a sequential version of contract design problem with cumulative utility maximization? I am thinking of a case where the principle also is deemed responsible of its actions in the contract design and there is a dynamics interaction between the agent and the principle.
3) Is there a specific reason to pick a mother neural network to models the bias term? It'd be more straightforward to write the bias term  as a linear function of activation pattern $ b^{L+1}(\mathbf{f}) = \sum_{l=1}^{L} \mathbf{r_l(\mathbf {f})}^T \mathbf{b}_l $.
4) What are some real world applications where the the offline learning with previously collected data could be feasible and reasonable? Are these type of data readily available in practice?
5) Does it require a significantly new treatment to extend this to multi-agent (single principle) case?

**Limitations:**

Limitations, societal considerations and future works are all discussed.

---

> ### Author Rebuttal · Authors · 2023-08-09
>
> Thanks for the thought-provoking review, which prompts us to think deeper about the applicability and extension of our work.
>
> > (Q1) _`How easy is it to extend your method to contracts with mixed payment-penalty structure?`_
>
> For this, we can consider the case $\mathbf{f}\ge-c$, for a positive constant $c$  (with no limitation on how much the agent can pay the principal, there is a trivial optimal solution, charging the agent the entire expected social welfare). Everything then goes through immediately, with contracts sampled $\mathbf{f}\ge-c$, and the non-negativity constraints in the inference replaced with $\mathbf{f}\ge-c$.
>
> &nbsp;
> > (Q2) _`Is there a sequential version of the contract design problem with cumulative utility maximization?`_
>
> This is an interesting direction, and sequential versions of the contract design problem are a classic topic in the economics literature; e.g., the seminal paper [1], see also [2]. Studying these models from a computational direction is an interesting direction for future work.
>
> In the future, if (deep) (multi-agent) reinforcement learning is explored to study more comprehensive or larger scale sequential problems, we believe the proposed DeLU architecture can contribute by approximating the Q function to estimate the long-term value the principal can expect with a contract. Such a Q function is discontinuous, with the agent's response a sequence of discrete actions.
>
> [1] Holmstrom, B. and Milgrom, P., 1987. Aggregation and linearity in the provision of intertemporal incentives. Econometrica, pp.303-328.
>
> [2] Zhang, H. and Zenios, S., 2008. A dynamic principal-agent model with hidden information: Sequential optimality through truthful state revelation. Operations Research, 56(3), pp.681-696.
>
> &nbsp;
> > (Q3) `Is there a specific reason to pick a mother neural network, instead of a linear function, to model the bias term?`
>
> The reason for using a second network is that the bias does not always depend linearly on the activation pattern. Here is an example to illustrate this. There are two outcomes with values $\mathbf{v}=[20,1]$,  four actions with costs $\mathbf{c}=[1.0,2.1,2.3,4.7]$, and the action-outcome transition kernel is$$P=\begin{bmatrix}0.211&0.789\\\\0.398&0.602\\\\0.430&0.570\\\\0.684&0.316\end{bmatrix}.$$
> Suppose we consider linear contracts, where $\mathbf{f}=\alpha\mathbf{v},\alpha>0$. Then the principal's utility function for different contracts is$$u^p(\alpha)=\begin{cases}-5\alpha+5&0.2<\alpha<0.3\\\\-8.57\alpha+8.57&0.3<\alpha<0.4\\\\-9.17\alpha+9.17&0.4<\alpha<0.5\\\\-14\alpha+14&\alpha>0.5\\\\ \end{cases}.$$Suppose that we have a 2-dimensional activation pattern, and the linear function converting activation patterns to the bias has parameters $[b_1,b_2]$. Then the bias for each of the four pieces would be $0$, $b_1$, $b_2$, and $b_1+b_2$, respectively. The difference between each piece's bias needs to model the discontinuity at contract parameter $\alpha=0.3,0.4,0.5$, but this is impossible with this linear model. To see this, we first assume that the piece $0.2<\alpha<0.3$ has bias 0. Then the differences of biases of the other 3 pieces would need to be 2.5, 2.86, and 5.28, which cannot be achieved with $b_1$, $b_2$, and $b_1+b_2$. It can be easily verified that the cases where other pieces have a bias of 0 are similar, demonstrating that a linear bias function cannot express the discontinuity. By contrast, appealing to a second network allows for non-linear dependency on activation, and can handle this problem.
>
> &nbsp;
> > (Q4) _`What are some real world applications where offline learning with previously collected data could be feasible and reasonable? Are these type of data readily available in practice?`_
>
> First of all, an application that motivates us is that of developing a tool for theoretical economists to study contract design. In economic theory, one typically assumes a model of a world environment (e.g., actions, technology, costs, rewards, outcomes), and looks to understand the optimal designs. In this setting, offline data is reasonable because we have a model of the world.
>
> In regard to real-world, practical applications:
>
> (1) One application comes from the _digital economy_, where we can expect to gain access to training sets as we see increasing automation of economic processes (e.g., it doesn't seem too unlikely that we will soon have contracts for LLM-style actors, for example working to plan the details of a vacation for a user with the possibility of contracting on outcomes) [3].
>
> (2) Another application comes from settings where we can expect to have access to a simulator of the behavior of  agents. In particular, a black-box simulator fits perfectly  with our set-up, where we only need access to the induced utility to the principal for different contract designs. These simulations may be for automated agents in the digital economy. Another interesting development is that attention is going these days to generative models as simulators of _human decision making and behavior_ (and in the future, likely firm behavior) [4].
>
> As these models are developed, methods to optimize on top of them will become important, and we expect our method to be useful in regard to contract design.
>
> [3] Horton, J.J., 2023. Large language models as simulated economic agents: What can we learn from homo silicus? (No. w31122). National Bureau of Economic Research
>
> [4] Park, J.S., O'Brien, J.C., Cai, C.J., Morris, M.R., Liang, P. and Bernstein, M.S., 2023. Generative agents: Interactive simulacra of human behavior. arXiv:2304.03442
>
> &nbsp;
> > (Q5) _`Does it require a significantly new treatment to extend this to multi-agent (single principal) case?`_
>
> Not a significantly new treatment, as long as we have access to the behavior model of the system of agents. This would need to resolve, for example, a question in regard to computing or observing equilibrium behavior, perhaps also involving tie-breaking across multiple possible equilibria.

---

> > ### Comment · Reviewer_z4Vn · 2023-08-22
> >
> > Thank you very much for addressing my questions as well as other other reviewers' concerns. Based on all the other reviews and authors' responses, I'm in favor of maintaining my score.

---

> > > ### Author Response · Authors · 2023-08-22
> > > **Thanks a lot for your response!**
> > >
> > > We would like to express our gratitude for the reviewer's valuable comments and inputs. We find the questions really helpful in improving the quality of our work. Specifically, we plan to make the following changes to our paper:
> > >
> > > - Discuss why we use another network to generate the bias of the last layer, instead of a linear transformation (around Line 204).
> > > - Provide an extensive discussion about the real-world applications of the proposed method, and whether previously collected data is feasible in these applications (in a future work section).
> > > - Discuss possible extensions of our work to multi-agent scenarios, sequential contract design, and mixed payment-penalty settings in the Limitation section.

---

### Author Rebuttal · Authors · 2023-08-09

We would like to express our sincere gratitude to the reviewers for providing exceptionally high-quality and insightful reviews. Your thoughtful insights and valuable suggestions have significantly enriched our work. Thank you for your time, effort, and commitment, and we look forward to addressing your further comments during the discussion period.

Here in the general response, we show the following figures in the attached PDF:

- Figure 1. Optimality (normalized principal utility) of (1) _the best sample in the training dataset_, (2) DeLU, (3) ReLU, and (4) a direct LP solver (“Oracle LP"), for contract-design problems with increasing sizes.

- Figure 2: The performance of the DeLU network trained with _different numbers of training samples (log scale)_ on three sizes of problems (# actions, # outcomes). The median performance as well as the first and third quartile (shaded area) of 5 combinations of $(\alpha_p, \beta_p)$ are shown.

---

### Decision · Program_Chairs · 2023-09-21

**Decision:**

Accept (poster)

**Comment:**

This paper provides a new approach for the contract design problem based on (1) supervised learning approach for learning utility, (2) a new DeLU network architecture for better approximating the discontinuous utility, and (3) efficient gradient-based inference to find approximate optimal contract on a learned DeLU network. The approach shows advantages in runtime and solving larger-scale problems over existing methods such as Linear Programming on the exact utility.

The reviewers are unanimously positive about the paper after rebuttal. Therefore, I recommend acceptance.